# Geographic and Ecological Diversity of Green Sulfur Bacteria in Hot Spring Mat Communities

**DOI:** 10.3390/microorganisms11122921

**Published:** 2023-12-04

**Authors:** Donna L. Bedard, Greta Van Slyke, Ulrich Nübel, Mary M. Bateson, Sue Brumfield, Yong Jun An, Eric D. Becraft, Jason M. Wood, Vera Thiel, David M. Ward

**Affiliations:** 1Department of Biological Sciences, Rensselaer Polytechnic Institute, Troy, NY 12180, USA; donna.l.bedard@gmail.com (D.L.B.); gretavanslyke@gmail.com (G.V.S.);; 2Department of Land Resources and Environmental Sciences, Montana State University, Bozeman, MT 59717, USA; ulrich.nuebel@dsmz.de (U.N.); marymbateson@gmail.com (M.M.B.); ebecraft@gmail.com (E.D.B.); jasonmw@uic.edu (J.M.W.); 3Leibniz-Institute DSMZ German Collection of Microorganisms and Cell Cultures, 38124 Braunschweig, Germany; vera.thiel@dsmz.de; 4Department of Plant Sciences and Plant Pathology, Montana State University, Bozeman, MT 59717, USA; sbrumfield@montana.edu; 5Department of Biology, University of North Alabama, Florence, AL 35632, USA; 6Research Informatics Core, University of Illinois at Chicago, Chicago, IL 60607, USA

**Keywords:** *Chlorobaculum tepidum*, photosynthesis, chlorosomes, bacteriochlorophyll *c*, thermophile, ecological species, biogeographic distribution

## Abstract

Three strains of thermophilic green sulfur bacteria (GSB) are known; all are from microbial mats in hot springs in Rotorua, New Zealand (NZ) and belong to the species *Chlorobaculum tepidum*. Here, we describe diverse populations of GSB inhabiting Travel Lodge Spring (TLS) (NZ) and hot springs ranging from 36.1 °C to 51.1 °C in the Republic of the Philippines (PHL) and Yellowstone National Park (YNP), Wyoming, USA. Using targeted amplification and restriction fragment length polymorphism analysis, GSB 16S rRNA sequences were detected in mats in TLS, one PHL site, and three regions of YNP. GSB enrichments from YNP and PHL mats contained small, green, nonmotile rods possessing chlorosomes, chlorobactene, and bacteriochlorophyll *c*. Partial 16S rRNA gene sequences from YNP, NZ, and PHL mats and enrichments from YNP and PHL samples formed distinct phylogenetic clades, suggesting geographic isolation, and were associated with samples differing in temperature and pH, suggesting adaptations to these parameters. Sequences from enrichments and corresponding mats formed clades that were sometimes distinct, increasing the diversity detected. Sequence differences, monophyly, distribution patterns, and evolutionary simulation modeling support our discovery of at least four new putative moderately thermophilic *Chlorobaculum* species that grew rapidly at 40 °C to 44 °C.

## 1. Introduction

Green sulfur bacteria (GSB) are obligately anaerobic, anoxygenic photoautotrophic bacteria that oxidize hydrogen sulfide to elemental sulfur and sulfate while fixing inorganic carbon. They are important for several reasons. First, they are unique among photosynthetic organisms in their use of the reverse tricarboxylic acid cycle for carbon fixation [1]. This pathway was discovered in GSB [2] and is found only in members of a few other taxa, including *Epsilonproteobacteria* (*Pseudomonadata*, *Bacteria*), *Thermoproteus* spp. (*Thermoproteoa*, *Archaea*), and members of the deeply branching *Hydrogenobacter* and *Aquifex* (both *Aquificota*, *Bacteria*) [3,4,5,6]. Second, GSB arguably provide one of the best models of primordial photosynthetic bacteria [7,8]. GSB possess a homodimeric Type-1 photosynthetic reaction center, which is generally accepted as most closely resembling the reaction centers of the earliest photosynthetic organisms [7,9]. Finally, although anoxic environments in which GSB can form mass accumulations are not common on modern Earth, they may have been more widespread earlier in Earth’s history [10,11]. Therefore, modern communities that include thermophilic GSB may offer novel insights into ancestral photosynthetic communities.

On the basis of 16S rRNA and Fenna-Matthews-Olson (FMO) protein gene sequences [12,13] and genome-wide conserved indel patterns [14], GSB comprise a monophyletic group of bacteria, which contains only four genera: *Chlorobium* (*Chl*.), *Chlorobaculum* (*Cba*.), *Prosthecochloris*, and *Chloroherpeton*. The first three genera belong to the family *Chlorobiaceae*, and most members of this family can oxidize sulfide to sulfate, whereas *Chloroherpeton* lacks the *dsr* genes and cannot oxidize sulfur to sulfate [15]. Many mesophilic strains of GSB have been isolated from various freshwater and marine environments, but the upper temperature for growth of these strains is around 35 °C [16,17,18]. Thermophilic GSB have previously been found only in hot springs in the Rotorua area of New Zealand (NZ) [19]. Three strains have been isolated: *Cba*. *tepidum*, strains TLS^T^ (Travel Lodge Stream, type strain), S. PTE.-1 (Sulphur Pointe-1), and GVP (Government Vent Pool). They were cultivated from thermal springs with temperatures of 40 to 55 °C, pH 4.3 to 6.2, and sulfide concentrations ranging from 0.2 to 1.8 mM [19,20]. It was proposed that thermophilic GSB could exist only within that pH and sulfide concentration range because low pH would prevent the growth of *Chloroflexus* (*Chloroflexia*, *Chloroflexota*) and the combination of high temperature, low pH, and high sulfide would prevent the growth of cyanobacteria [19,21,22].

In 1977, Castenholz surveyed a large number of hot springs in the Mammoth Terraces region of Yellowstone National Park (YNP), Wyoming, USA [22]. He reported phototrophic mats composed of *Chloroflexus* spp. and “a *Chlorobium*-like unicell” in several of the sulfidic springs there (pH 6.2 to 6.8, 51 to 66 °C, and sulfide concentrations ranging from 32 to 108 µM). This report of “*Chlorobium*-like” cells was based solely on light microscopy and was never confirmed. Mats at these sites are composed of members of the *Chloroflexota* and sometimes *Thermochromatium tepidum*, a purple sulfur bacterium (*Gammaproteobacteria*, *Pseudomonadota*) [23,24]. No thermal springs with conditions exactly like those in Rotorua hot springs are known in YNP. For instance, none are known with sulfide concentrations in the mM range, and there are relatively few sulfidic thermal springs in the pH range of 4.5 to 6.2 (Kirk Nordstrom, USGS, 2000, personal communication).

Partial 16S rRNA sequences with approximately 89% sequence identity values to GSB reference sequences were detected in 60 to 65 °C cyanobacterial mats of Octopus Spring and Mushroom Spring, two well-studied alkaline thermal systems in YNP [25]. However, these were subsequently shown in metagenomic analyses to be associated with an aerobic, anoxygenic photoheterotroph member of the *Chlorobiota*, *Candidatus* Thermochlorobacter aerophilum [26], instead of anaerobic, anoxygenic GSB.

In this study, we present the results of our search for thermophilic GSB residing in mats at three different geographic locations: (i) Travel Lodge Stream in Rotorua, NZ, from which *Cba*. *tepidum* was cultivated; (ii) a hot spring in the town of Lipayo on Negros Island in the central Philippines (PHL); and (iii) some mildly acidic to nearly neutral sulfidic hot springs of YNP (Figure 1A,B). Selective enrichment and targeted 16S rRNA primers were used to search for GSB diversity. Restriction fragment length polymorphism (RFLP) of 16S rRNA genes was used to analyze the eubacterial diversity in a GSB mat community in Lipayo thermal spring, PHL. In YNP, we focused our search on the Gibbon Geyser Basin, Mud Volcano, and Mammoth Hot Springs regions (Figure 1B) because they include hot springs most similar to those around Rotorua, with pH values between 4.5 and 6.9, temperatures in the moderately thermophilic range (35 to 50 °C), and low to moderate sulfide concentrations.

Here we report the discovery of a diverse range of putative moderately thermophilic GSB from the genus *Chlorobaculum* in hot spring mats associated with these distinct thermal regions. We consider the taxonomic implications of our results in terms of (i) phylogenetic analysis and the degree of sequence divergence, and (ii) a theory-based evolutionary ecological simulation model of species and speciation, Ecotype Simulation [27]. Our results expand the known diversity of *Chlorobaculum* spp. inhabiting thermal environments and provide evidence of diversification due to both geographic isolation and ecological specialization.

## 2. Materials and Methods

### 2.1. Bacterial Strains

*Cba*. *tepidum* strains TLS^T^ and S. PTE-1 were obtained from Professor Michael Madigan (Southern Illinois University, Carbondale, IL, USA) and were cultivated as described [20]. All other GSB strains used for testing primer specificity (see Section 2.9) were obtained from the German Collection of Microorganisms and Cell Cultures (DSMZ, Braunschweig, Germany) and cultivated according to the recommendations of the supplier.

### 2.2. Sample Collection

Three different hot spring systems from different continents were sampled for this study: Travel Lodge Spring (TLS) in Rotorua, NZ; Lipayo hot springs, PHL; and several different hot springs in Yellowstone National Park, WY, USA. The samples collected in this study and their environmental characteristics are described in Table 1 and Appendix A. The pH of the water overlying the mats in hot springs was measured with a Corning Model 313 portable pH meter with built-in temperature compensation (Burlington, MA, USA). Temperatures were measured with a Corning Model 309 portable temperature meter (Burlington, MA, USA). Samples for enrichment were collected aseptically using forceps, spatulas, or Eppendorf pipets and transferred to sterile screw-cap microfuge tubes or cryovials, which were then filled completely with spring water from the collection site. The sample from NZ was collected on 9 September 1986, and was the same collection that led to the cultivation of *Cba*. *tepidum* [20]; no enrichment was made in our current study. Samples from YNP and PHL were all collected between 2000 and 2002, kept cool during transport, and then stored at 4 °C for up to 3 weeks before enrichment. Samples for DNA extraction were immediately frozen on dry ice and then stored at −20 °C, or in the case of NZ, at −70 °C. Samples for sulfide analysis were preserved by adding a 2% zinc acetate solution (1:1); colorimetric analysis was performed later [28].

### 2.3. Light Microscopy

Phase contrast and autofluorescence microscopy were performed using an Axioplan microscope equipped with a NEOFLUAR objective in combination with fluorescence filter set 15, which restricted green excitation to 546 nm and red emission to wavelengths > 590 nm (Carl Zeiss Inc., Oberkochen, Germany).

### 2.4. Enrichment Conditions

We used a modification of Pfennig’s medium [29,30] buffered with 15 mM enzyme-grade MOPS and adjusted to pH 6.9, described as Maintenance Medium [20], for all enrichment and transfer cultures of GSB. This medium contained 1 mM sulfide. Full instructions for the enrichment and growth medium are given in Appendix A. In some cases, several different temperatures were tried (see notes to Table 1). We tested all enrichments with and without 4 mM sodium thiosulfate in addition to sulfide. As observed for *Cba. tepidum* strain TLS [20], the best results were obtained with the thiosulfate supplement, except for BS3.1 and BS3.2 enrichments. In those two cultures, thiosulfate encouraged the growth of purple sulfur bacteria as well as GSB. All media were prepared aseptically and anoxically using a sterile stream of N_2_ gas. The same method was used to maintain anoxic conditions during inoculations and transfers. Enrichments were incubated in filled, 17 mL screw-cap tubes in water baths or incubators at the temperatures indicated in Table 1 for up to three weeks or until good growth occurred. The light source was a 40 W incandescent light bulb in all cases. The incubation temperature used for most enrichments was close to or slightly lower than the temperature at which the sample was collected. The Mud Volcano Site 6 (MV6; see Table 1) enrichment was 4 °C higher than the mat temperature. The Mammoth Hot Springs Angel Terrace Spring (AT, see Table 1) enrichment was only tested at temperatures of 55 °C and lower, even though the mat was 67.4 °C.

### 2.5. Isolation of GSB Strain MV4-Y

In 2011, GSB enrichments from site MV4 that had been grown at 44 °C, then adjusted to 10% DMSO for freezing and stored frozen at −70 °C, were revived by growing liquid cultures at 44 °C in the same medium used for enrichments and growth, in this case with thiosulfate (MV4-44). These were used to inoculate, in triplicate, deep soft-agar (1.7% *w*/*v*) tubes using a modified version of the isolation technique of Loeffler et al. [31] in order to obtain single colonies. The tubes were incubated in a water bath at 44 °C under incandescent light until individual green colonies appeared. Single dark-green colonies from the first soft-agar deep dilution series were obtained for enrichment MV4-44 and inoculated into liquid media as above. Once the liquid culture had grown, it was used to inoculate additional dilution series in deep-agar tubes until a pure culture, MV4-Y, was obtained.

### 2.6. Chlorophyll and Carotenoid Analysis

Absorbance spectra of GSB enrichments were determined for whole cells immediately after enrichment. Cell pellets were obtained using a microcentrifuge, resuspended in 60% sucrose, and homogenized in a glass homogenizer. Scans from 400 to 1000 nm were recorded on a Hitachi U-2000 dual-beam scanning spectrophotometer (Hitachi USA, San Jose, CA, USA).

### 2.7. Transmission Electron Microscopy

GSB cells enriched on Maintenance Medium plus thiosulfate were concentrated into a pellet, which was washed twice with sodium-potassium phosphate buffer (23 mM NaH_2_PO_4_·H_2_O, 77 mM K_2_HPO_4_, pH 7.3) and then adjusted to 3% (*v*/*v*) glutaraldehyde. The fixed samples were kept at 2 to 5 °C until they were prepared for transmission electron microscopy as described [32]. Thin sections were examined at 100 kV with a Leo 912 electron microscope (Carl Zeiss, San Diego, CA, USA).

### 2.8. DNA Extraction and PCR Amplification

DNA was extracted from the frozen TLS sample collected in 1986 and from mat samples collected in 2000–2002, as soon as possible after collection, using a bead-beating protocol [33]. PCR reactions (25 or 50 μL) were performed according to the recommendations of the polymerase manufacturer. Three concentrations (undiluted and 10^−1^ and 10^−2^ dilutions) of DNA extract from each mat sample were used as templates for PCR amplification to minimize PCR inhibition from environmental contaminants. It was not necessary to extract DNA from axenic GSB strains or from freshly grown enrichment samples prior to amplification. For those samples, the cells were washed twice with phosphate-buffered saline, resuspended in 10 mM Tris-HCl, 1 mM EDTA, pH 8, and used directly as a template. Promega Taq Polymerase (Promega, Madison, WI) or Master Taq Polymerase (Eppendorf, Enfield, CT, USA) were used for amplifications. Nucleotides were purchased from Promega (Promega, Madison, WI) or Fisher Scientific (Waltham, MA, USA). Positive (*Cba. tepidum* TLS^T^ cells) and negative (no template) controls were included for each PCR trial. Results were considered valid only if both controls were successful and only after confirmation by an independent replicate amplification. PCR conditions were optimized along with the primer design (see below) and then used for amplification from mat and enrichment samples.

### 2.9. GSB Primer Design and PCR Optimization

Primers used in this study are listed in Table 2. A forward primer, GSB600F, that specifically targets GSB was designed: 5′ GGG GGT TAA ATC CAT GTG was paired with the universal primer 1392R as a reverse primer (Table 2). This primer pair, which spans 16S rRNA hypervariable regions V4, V5, and V6 [34], was a perfect match for the 16S rRNA sequences of 30 axenic GSB strains that were in GenBank in April 2000; specificity was further confirmed by a BLAST search against the full NCBI nucleotide collection nr/nt database containing 96 million sequences in April 2023. These primers were tested for specificity against eight axenic GSB strains, representing three of the four GSB genera, and six strains unrelated to GSB. The GSB strains used were: *Cba. tepidum* TLS^T^, *Cba. tepidum* S. PTE.1, *Cba. luteolum* DSM 273^T^, *Chl*. *limicola* DSM 245^T^, *Chl. phaeobacteroides* DSM 266^T^, *Chl. phaeovibrioides* DSM 269^T^, *Prosthecochloris* (*Ptc*.) *aestuarii* DSM 271^T^, and *Ptc. vibrioformis* DSM 260^T^. The unrelated strains tested were *Escherichia* (*E.*) *coli*, *Bacillus subtilis* DSM 402, *Cytophaga johnsonae* DSM 2064, *Desulfobacterium vacuolatum* DSM 3385, *Rhodobacter capsulatus*, *Spirulina* sp. PCC 6313, and *Thiomicrospira thyasirae*. At an annealing temperature of 57 °C, all GSB gave strong PCR products of the expected length, and no products were detected from the unrelated bacteria. The optimal PCR conditions were as follows: initial denaturation at 94 °C for 2 min; 30 cycles of denaturation at 94 °C for 45 s; annealing at 57 °C for 60 s; extension at 72 °C for 60 s; final extension at 72 °C for 7 min; hold indefinitely at 4 °C. In some cases, GSB sequences could not be detected in mat DNA by direct PCR with the GSB600F/1392R primer set. These samples were retested by nested PCR using the primer set 27F/1492R (Table 2) and an annealing temperature of 45 °C for 30 cycles. This was followed by a conventional hot-start PCR with the GSB600F/1392R primers. Even with this procedure, it was not possible to amplify GSB sequences from several sites that nevertheless yielded dense GSB enrichments (see Table 1). However, GSB sequences were successfully amplified from those sites with the GSB-specific primer set GSB.619F/GSB1144R ([35], Table 2). It is not clear why the GSB 600F/1392R primer set did not yield GSB sequences from some mats. We speculate that it may be because of the competition of DNA from non-GSB bacteria present at the site, which could also bind to the universal 1392R primer. Therefore, all samples that did not yield PCR products with the GSB600F/1392R primer set were retested with the set of two GSB-specific primers (GSB619F/GSB1144R).

### 2.10. PCR of Eubacterial 16S rRNA Genes

We chose to sample the Eubacterial diversity in the Lipayo-5 mat more broadly because of the high density of white streamers, presumably from sulfide and/or sulfur-oxidizing organisms, which was not seen at any of the other sites. In 2002, we used the Eubacterial primer 27F and the universal primer 1492R (Table 2) to amplify the 16S rRNA genes from DNA extracts of samples collected at the Lipayo hot spring and create clone libraries of nearly full-length 16S rRNA genes (see Section 2.11). These were analyzed by RFLP [39,40] using the restriction enzymes MSP I and Hha I (0.1 unit each overnight at 37 °C) and then visualized on a 2% agarose gel. Representatives of each RFLP type were sequenced. These sequences were initially tentatively identified using BLAST [41,42] in 2002, and subsequently, in 2023, taxonomically identified by BLAST restricted to type strains, the Ribosomal Data Base classifier [43], and the List of Prokaryotic Names with Standing in Nomenclature [44,45,46] and recorded in Appendix A. The same method was used to obtain nearly full-length sequences of 16S rRNA from the GSB enrichment derived from the Lipayo-5 mat.

### 2.11. Cloning and Sequencing

PCR products were cloned in *E. coli* using the Topo TA cloning kit (Invitrogen, Waltham, MA, USA). The plasmids with inserts were extracted using the QIAprep Spin Miniprep Kit (Qiagen, Germantown, MD, USA) and sequenced using the appropriate targeted amplification primers. Sequencing reactions were prepared with Applied Biosystems Prism^®^ Big Dye™ Terminator Cycle Sequencing Ready Reaction Kit V. 3.0 or 3.1 (Waltham, MA, USA). Sequences were analyzed on an Applied Biosystems 3700 DNA Analyzer (Waltham, MA, USA) and edited and aligned using The Wisconsin Package, version 10.0—Unix (1982–1998) (Genetics Computer Group, Inc., Princeton, NJ, USA; Oxford Molecular Group, Inc., Cambridge, UK). The sequences were compared with the NR database in GenBank using the BLAST tool [41,42]. Sequences have been deposited in GenBank and assigned accession PopSet: 40556720 and accession numbers OQ518241-OQ518253, OQ518371, OQ519263-OQ519273, OQ726108-OQ726109, and OQ913398-OQ913416.

### 2.12. Phylogenetic Analysis

In 2023, the sequences obtained from 2000 to 2002 were taxonomically identified by comparison to known sequences in NCBI nr/nt databases by BLAST search [41,42] and phylogenetic analysis using the ARB software package [47]. Imported sequences were aligned automatically using the pt_server database and manually corrected based on secondary structure information. Initial phylogenetic affiliations were obtained by adding the aligned sequences to the tree_SSURefNR99_1200_slv_138 tree backbone implemented in SILVA (SSU Ref. NR 139, released December 2019) [48]. Phylogenetic trees were generated based on the maximum likelihood method using the PhyML software included in the ARB package [47,49]. The inferred confidence was based on 100 bootstrap replicates, and only values of >50 are shown in phylogenetic trees. Only sequences with a length of ≥1240 nt were used for phylogenetic calculations. Shorter sequences (<760 nt) were added to the tree using the ARB parsimony method without changing the tree topology. The validity of subclades containing sequences ≤1240 nt could not be confirmed by bootstrap analysis because of the use of different methods of tree construction. Sequence identity values of 16S rRNA genes were obtained by the generation of a distance matrix based on the Neighbor Joining method [ARB_DIST] using the similarity correction implemented in the ARB software package [47].

### 2.13. Ecotype Simulation: Demarcation of Putative Ecotypes

We used an evolutionary ecology simulation model, Ecotype Simulation (ES) [27], which is based on the Stable Ecotype Model of species and speciation [50], to predict ecologically distinct (ecotype) or geographically isolated (geotype) clusters in the phylogeny. The Stable Ecotype Model assumes that ecotypes (or geotypes) arise during the evolution of a lineage through a progression of periodic selection events and occasional ecotype formation events (or physical isolation events). A phylogeny is first summarized as a binning curve, which is the number of sequence clusters as a function of the sequence relatedness used to define clusters. For example, when each sequence is deemed unique, the number of clusters equals the total number of sequences, but as sequences are clustered at lower relatedness values, clusters begin to contain more than a single sequence, and the number of clusters drops, ultimately to a single cluster containing all sequences. The binning curve is used as a standard for estimating the rate of periodic selection, the net ecotype (or geotype) formation rate, and the number of ecotype/geotype populations. These predicted parameter values are optimized in repeated Monte Carlo simulations of the evolution of the lineage, with the order of events and times at which they occur determined randomly, seeking a combination of values that best matches the binning curve, utilizing the downhill simplex method of Nelder and Mead [51]. Approximate 95% confidence intervals for each optimized parameter value are similarly calculated by allowing its value to vary while holding other parameter values static during repeated simulations. These optimized parameter values are used to recursively progress through each sub-clade of the phylogeny to find sequences and sequence clusters for which the maximum likelihood solution for the number of ecotypes (or geotypes) in the clade is equal to one.

It is important to note that the ES analysis is based only on phylogenetic information and does not include any environmental data, so that the basis for the distinctions between ecotypes and/or geotypes depends on subsequent correlative analyses. In this study, an improved version of ES, termed ES2, which can analyze a much larger number of and longer sequences faster, was used (Wood, J.M., Becraft, E.D., Krizanc, D., Cohan, F.M., Ward, D.M. 2020, *BioRxiv*). Directions for download and instructions for the use of the algorithm are freely available at https://github.com/sandain/ecosim (accessed on 1 March 2023). As will be discussed below, both ES1 and ES2 have been used successfully in predicting ecotypes and geotypes [52,53,54] from large sequence datasets obtained from hot spring microbial mats.

### 2.14. Genome Comparison

Full genome information of strain MV4-Y, which was sequenced by the laboratory of Prof. Donald A. Bryant (NCBI acc. no. CP104202.1), was compared to other *Chlorobaculum* spp. type strain genomes for average nucleotide identity (ANI) values using the ANI calculator found at the website of the Environmental Microbial Genomics Laboratory at Georgia Institute of Technology (http://enve-omics.ce.gatech.edu/ani/, accessed on 20 November 2023) with standard settings [55,56].

## 3. Results

### 3.1. Site and Phototrophic Mat Descriptions

Table 1 and Appendix A give the date of collection, the temperature, and the pH for each sample, as well as sulfide concentrations for some samples. Figure 2 and Figure 3 show many of the sites described below.

#### 3.1.1. Travel Lodge Spring, Rotorua, NZ

A sample (designated TLS) was collected on 9 September 1986 from a 44.5 °C, pH 5.8 site in the sulfidic (280 to 1850 µM) Travel Lodge Stream in Rotorua (Figure 2A), the location of which is described in Castenholz et al. [19]; it was frozen on dry ice, then stored at –70 °C until analysis in 2000. As shown in Figure 2B, a thin, ropey green accumulation, assumed to contain GSB, was present above the blackened sediment.

#### 3.1.2. Lipayo Hot Spring, Negros Island, PHL

The hot spring that we call “Lipayo” is a sulfidic thermal stream within the town of Lipayo, Negros Oriental, PHL (Figure 2C). The stream originates at a spring within an active geothermal site near a dormant volcano field and amid a classic old-growth rainforest. It is in a remote location, which we accessed by hiking in with a guide provided by the Southern Negros Geothermal Production Field of the PNOC Energy Development Corporation. At the source, the Lipayo hot spring (Figure 2C) was 57.7 °C and pH 4.47. Over a distance of 15 m downstream from the source pool, the temperature and pH of the stream ranged from 54.9 to 36.1 °C and from pH 3.3 to 6.0. At and below about 49 °C, the stream had a heavy growth of white fibrous streamers typical of sulfide/sulfur-oxidizing microorganisms and visible elemental sulfur precipitates. The air also had a strong smell of sulfide.

The sample, designated Lipayo-5 (Lip-5), was collected on 22 January 2002, about 15 m downstream of the source near the shore, just below where a small non-thermal stream joined the thermal stream, reducing the temperature to 36.1 °C and raising the pH to 5.1 to 6. At this site, the stream took on a yellowish-green color at the surface close to the shore (Figure 2D); the sulfide concentration was 40 µM. The samples contained grayish-white gelatinous material as well as pale yellow-green growth.

#### 3.1.3. Yellowstone National Park, WY, USA

##### Gibbon Geyser Basin

An unnamed spring, which we will refer to here as “Big Spring” (BS) (Figure 3A), is a large, spring-fed pool in the Gibbon Geyser Basin, located east of the Artists Paint Pots Thermal Area. In 2000, we sampled at three locations at the southwestern end of the pool within ~10 m of each other (sites BS1, BS2, and BS3). Site BS1 (41.5 °C, pH 6.4) harbored a bright, green mat lying on muddy sediment at the juncture of an inlet and the pool (sample BS1-00) (Figure 3A,B). In August 2001, site BS1 was still much the same, but the temperature had increased to 51.6 °C; sample BS1-01 was collected. Site BS2 (51.7 °C, pH 4.6; not shown in Figure 3) was located 0.5 m to the east of, and was similar in appearance to, site BS1: a yellowish-green mat was lying on mud with a thin white film (possibly sulfur) on top. The water depth was 7 to 10 cm; however, we noticed that the water level was periodically rising and falling within minutes. When the water level rose, the pH rose from 4.6 to 6.4, and the temperature dropped from 51.7 to 46 °C (see Table 1).

Site BS3 was located in a small inlet of “Big Spring” in the center of the southern end of the pool (Figure 3A). Here, the water depth was about 25 cm. On 21 June 2000, there was a pinkish-purple and yellowish-green phototrophic mat about 7.5 × 10 cm in size floating on the surface of the pool. There was an acidic inlet immediately to the west of the mat at site BS3 and an alkaline thermal vent (71.8 to 72.2 °C, pH 8.6 at the center) 5 cm to the east of the mat; hence, the site BS3 mat was located within a temperature and pH gradient. The mat sample, designated BS3.1, was collected at 35.6 °C and pH 6.4 on that date. By 29 June 2000, the mat had grown to 10 × 12.5 cm. The temperature and pH of the mat ranged from 44.4 °C, pH 6.7 closest to the thermal vent, to 37.7 °C, pH 6.5 furthest from the vent. Samples were collected at 44.4 °C from the loose top (T) mat (sample BS3.2T) and from a deep-green middle (M) layer beneath it (sample BS3.2M). Below this was a bottom (B) layer of white filaments and black sediment (sample BS3.2B). This mat was no longer present when we visited the site in August 2001.

The BS1 and BS2 mat samples collected in June of 2000 were examined by phase-contrast microscopy shortly after collection. Both contained abundant small ovoid cells and objects that appeared to be external sulfur globules. There were also a few diatom frustules and red-fluorescent *Synechococcus* spp. cells. The samples from sites BS3.1 and BS3.2 also contained apparent sulfur globules, and their numbers increased with depth to the point that the deepest samples appeared milky. All three layers (T, M, and B) of the site BS3.2 sample had many different types of microorganisms, including diatom frustules. Many filamentous cyanobacteria were present in the surface layer of the mat (sample BS3.2T). The deep green layer (sample BS3.2M) and the white filament layer (sample BS3.2B) contained numerous small unicellular bacteria, many of which appeared to be associated with sulfur globules.

##### Mud Volcano Area

The GSB mats in the Mud Volcano (MV) area appeared as circles of loose, bright green or yellowish-green growth on fine sediment surrounding small bubbling springs (~25 mm diameter) that occurred in several pools of an effluent channel flowing northeast from Obsidian Pool (see [57] for location) (Figure 3C–F). Sites that we designated MV1 and MV2 (samples collected in 2000) were located about 50 m downstream of Obsidian Pool in a side pool (approximately 65 × 45 cm) of the Obsidian Pool effluent channel (Figure 3C). The mat at site MV1 grew in a circle around a bubbling spring with hot water flowing out at 70 °C. The water overlying the mat was only a few mm deep, and its temperature was 49 to 54 °C. Site MV2 was similar to site MV1, but the water temperature was lower (Table 1). Samples from sites MV4 and MV5 were collected from the same side pool in 2001. At this time, there was a single yellowish-green mat spreading over the entire side pool (Figure 3F). The water depth was 5 to 15 cm. Figure 3D shows a typical GSB bubbling spring from the Mud Volcano area. This particular photo shows the site MV3 mat (collected in 2001) (approximately 25 cm × 35 cm), which was upstream of the side pool (see Figure 3C). The water depth was 1 cm except for the vent, which was approximately 6 cm deep. Figure 3E shows a pool upstream of the side pool (see Figure 3C) containing many bubbling springs in very shallow water (approximately 1 to 2 cm).

The mat circling the large bubbling spring was designated site MV6 (collected in 2002) (Figure 3C,E). All of these sites had pHs within the range of 5.4 to 5.9. The bubbling springs were ephemeral and did not appear in precisely the same areas over the period 2000–2002. They also varied in temperature, ranging from 34 to 54 °C. When we first sampled the side pool in June of 2000, it contained several discrete bubbling springs, including sites MV1 and MV2. The following two years, a yellowish-green mat extended over the entire bottom of the pool (Figure 3F). The underlying silty sediment appeared black, possibly from metal sulfide deposits. The water covering the mat varied in depth at the different times sampled, ranging from 1 to 5 cm in 2000 and 2002 to 5 to 15 cm in 2001.

##### Mammoth Hot Springs

We sampled eleven mats in seven sulfidic hot springs in the Upper Terraces area of Mammoth Hot Springs. Most of the sites are briefly described in Appendix A, and the locations of many of the sampling sites are shown in Figure 1 of Castenholz [22]. Evidence for GSB sequences (see below) was obtained from only two of the Mammoth sites: Angel Terrace and Narrow Gauge 2, which are described in Table 1. The olive-green to orange mat sample from Angel Terrace (designated AT) was collected from a shallow pool at the top of the travertine terraces. This mat did not resemble the GSB mats at either Mud Volcano or “Big Spring”, which were all yellowish-green or bright green. The mat sample, designated NG2, was collected from a tiny spring emerging near the mid-point of the Narrow Gauge Spring ridge. The mat at this site was fibrous and dark olive-green and orange and was adjacent to white streamers. Fluorescence microscopy of this mat sample showed mostly short and long filaments and rods, as well as some red-fluorescing *Synechococcus*-like cells.

### 3.2. GSB Enrichments

Dense bright-green photoautotrophic enrichments were obtained from samples collected at site Lipayo-5, all Mud Volcano sites, and two of the three “Big Spring” sites at incubation temperatures near or slightly below those at which the mats occurred in nature (Table 1). A single exception is noted below. When enrichments for GSB were successful, dense, bright-green cultures were obtained after three to nine days of incubation. An enrichment obtained from the sample collected from site MV6 by dilution to extinction indicated that the original mat contained at least 2 × 10^7^ GSB cells mL^–1^. Furthermore, this enrichment was obtained at 44 °C, even though the temperature of the pool from which MV6 was collected was only 37.4 to 40 °C. The enrichments were readily subcultured. Stocks were adjusted to 10% (*v*/*v*) DMSO and remained viable after storage at –70 °C for up to two years (and in several cases for approximately 10 years).

Attempts to culture GSB from samples collected at Mammoth Hot Springs failed, even though 16S rRNA sequences characteristic of GSB were obtained from mats at two of these sites (see Table 1 and Molecular Studies in Section 3.6).

### 3.3. Absorption Spectra

Figure 4 shows the whole-cell absorbance spectra of GSB enrichments from sites MV1, BS3.2M, and Lipayo-5, which are typical of the spectra from all of our GSB enrichments. The in vivo absorption maximum for chlorobactene is 456 to 458 nm, and for Bchl *c* found in GSB, it is 745–756 nm [16] (as opposed to the Bchl *c* found in *Chloroflexus*, which has a maximum at 740 nm) [58,59]. The spectra of our enrichments are essentially identical to spectra taken from enrichments obtained from mat samples collected in Rotorua (Bchl *c* 753 nm) [19]. Thus, the whole-cell absorbance spectra support the inference that the organisms enriched from the hot springs of Lipayo-5 and from the Mud Volcano and “Big Spring” regions of YNP are predominantly GSB producing BChl *c*.

### 3.4. Transmission Electron Microscopy

Electron micrographs of thin sections of cells of GSB enrichments grown at 44 °C from sites MV4, BS1, and BS3.2M (Figure 5) revealed unicellular, non-flagellated, rod-shaped cells 0.24 to 0.28 μm × 0.5 to 1 μm. The cells exhibited ovoid or cigar-shaped inclusions at their periphery, typical of chlorosomes observed in *Cba. tepidum* TLS^T^ and other GSB [20,60]. The morphologies of the cells in the GSB enrichments were completely consistent with GSB belonging to the genera *Chlorobium* or *Chlorobaculum* [60].

### 3.5. Isolation of the Putative Moderately Thermophilic *Chlorobaculum* sp. Strain MV4-Y

We isolated strain MV4-Y from an enrichment derived from the MV4 mat by repeated serial dilution to extinction in deep soft-agar tubes incubated at 44 °C. Ultimately, three colonies were isolated from the triplicate 10^−6^ dilutions and were grown separately in liquid media. After three days of growth, the cultures were examined microscopically. At 100× magnification, the cells appeared green; at 400× magnification, they appeared as uniform, slightly oval to rod-shaped cells, and the cultures were deemed axenic. The axenic culture was designated strain MV4-Y.

Strain MV4-Y grew well at 44 °C, pH 6.9, on minimal medium (Maintenance Medium [20]). At his request, this axenic culture was sent to Prof. Donald A. Bryant at Pennsylvania State University. He and his lab subsequently sequenced its genome and deposited it in Genbank as NCBI Reference Sequence: NZ_CP104202.1. A whole genome comparison carried out in our study revealed 88.38% average nucleotide identity (ANI) to *Chlorobaculum tepidum* strain TLS (NC_002932.3) and 85.1% ANI to *Chlorobaculum limnaeum* strain DSM 1677 (NZ_CP017305).

### 3.6. Molecular Studies

#### 3.6.1. 16S rRNA Sequences Amplified from Mat and GSB Enrichment DNA

Table 1 presents outcomes for PCR amplification of sample DNA using the various primer pairs. GSB-specific PCR products of the expected size were recovered from samples from Travel Lodge Spring and site Lipayo-5, all mat samples in the Mud Volcano area, and all “Big Spring”samples collected in 2000. In addition, GSB-specific 16S rRNA sequences were amplified from DNA extracted from samples collected at Angel Terrace Spring and Narrow Gauge Spring, both in the Mammoth Hot Springs region (Table 1), but most sites in the Mammoth region tested negative (see Appendix A). PCR using the GSB600F/1392R primer set and DNA from all successful GSB enrichments produced amplicons of the correct size (Table 1).

#### 3.6.2. Sequencing Results

PCR amplicons obtained with GSB-specific primers from mat samples from Travel Lodge Spring and sites in the Mud Volcano Area, Gibbon Geyser Basin, and Mammoth Hot Springs, as well as from enrichments (Table 1), were cloned, and multiple clones for each amplicon were sequenced. In addition, the 16S rRNA genes of *Cba. tepidum* strain S. PTE-1, which was isolated from Sulphur Pointe-1 Spring in Rotorua [20], were amplified using 27F/1492R primers and sequenced.

A total of 16S rRNA genes amplified with Eubacterial primers from the Lipayo-5 mat sample were cloned and sorted using RFLP analysis, and 21 clones representing all RFLP types were sequenced (Appendix A). The dominant genotypes were Cyanobacteria (42.5%), while *Chlorobaculum* spp. represented the second most abundant RFLP type, comprising 23.3% of the obtained clones. Other constituents of the microbial community in the Lipayo-5 mat were identified as *Thiomonas* spp. (*Betaproteobacteria*, *Pseudomonadota*) and *Hydrogenobaculum* (*Aquificae*, *Aquificota*) based on sequence analysis.

##### Phylogenetic Analyses

The phylogeny shown in Figure 6 is based on 16S rRNA gene sequences detected by analysis of DNA from clones of both mat samples and enrichment cultures, as well as reference sequences. The GSB sequences that we recovered all represented members of *Chlorobaculum* spp., which form a clade to the exclusion of the *Chlorobium* sequences used as an outgroup in our analyses. Mat sequences from Travel Lodge Spring formed a single clade (designated clade TLS, green highlight in Figure 6) together with *Cba. tepidum* strain TLS^T^ and strain S. PTE.-1 sequences. The TLS mat sequences (designated TLS-Mat-A, -B, -C, -E, -L, -N-, and -R in Figure 6) exhibited microheterogeneity. Mat sequence A, which was identical to the sequences of *Cba. tepidum* strains TLS^T^ and S. PTE.-1, and mat sequence B, which differed from mat sequence A by only 1 nucleotide over 741 nt, were present in 10 copies each. Mat sequences L and N were present in 3 copies each, and mat sequences C, E, and R in 2 copies each; 20 other sequences were singletons. Sequences L, N, and C differed from Sequence A by only 3 nt. In all, 64% of the clones sequenced were represented by only 7 distinct sequences. These data suggest that the Travel Lodge Spring mat contains a single *Chlorobaculum* population exhibiting microheterogeneity, with the most abundant sequence being identical to the sequences of the two isolated *Cba. tepidum* strains.

Sequences recovered from the Lipayo-5 mat sample and enrichments therefrom (red highlight in Figure 6) formed two clades that were phylogenetically distinct from Travel Lodge Spring sequences and from each other. Lipayo-5 mat sequences formed a clade (designated LIP-MAT in Figure 6) near the root of the tree. However, the enrichment cultures from this mat sample contained sequences that formed a second clade (designated LIP-ENR in Figure 6), much closer to the TLS clade. This suggests that the Lipayo-5 mat sample contained two phylogenetically distinct populations of *Chlorobaculum* spp., one of which was favored by enrichment culture conditions. Also in this second clade was a sequence (strain i9-9) obtained from a GSB-dominated photobioreactor fed sulfide waste and operated at mesophilic temperature and neutral pH at the University of Queensland in Brisbane, Australia ([61], and Timothy Hurse, personal communication, 24 September 2022).

Sequences retrieved from Yellowstone samples also formed two distinct clades, one near the root of the tree (designated YNP-1-BS3.1/3.2-MAT) and a second clade at the top of the tree (designated YNP-2) (Figure 6). The YNP clades were phylogenetically distinct from the clades formed by Travel Lodge Spring and Lipayo-5 mat and Lipayo-5 enrichment sequences. The YNP-1-BS3.1/3.2 clade contains sequences retrieved from the mat of one “Big Spring” site that was sampled three weeks apart (sites BS3.1 and BS3.2, light-blue and medium-blue highlights in Figure 6). Two additional sequences, one from *Chlorobaculum* spp. strain C1, which was cultivated from Lake Waldsee, Germany [62,63], and another, strain 24CR, isolated from the Carmel River, CA [64], were also associated with the YNP-1-BS3.1/3.2-MAT clade. A further analysis of the clones in this clade showed that eight different sequence types were found within the two mat samples, all differing from each other in only 1–2 nucleotide positions (out of 737 nt) (designated BS3.1-00-MAT-A, -B, -C, -J, -N, and -11 and BS3.2M-00-MAT -11 and -20 in Figure 6). The dominant sequence type (represented by clone BS3.1-00-MAT-A) was found 22 times, while three other sequence types (BS3.1-00-MAT-B, -11, and -D) were found six, five, and two times, respectively; the rest were singletons. Clade YNP-2 is phylogenetically most closely related to *Chlorobaculum limnaeum* (98.0–98.8% 16S rRNA gene sequence identity values to the type strain DSM 1677^T^).

Figure 7 shows an expanded view of clade YNP-2, which contains the sequence from the isolate *Chlorobaculum* sp. strain MV4-Y as well as a variety of mat and enrichment samples obtained from “Big Spring,” Mud Volcano, and two Mammoth hot springs, Angel Terrace and Narrow Gauge, all located in Yellowstone National Park, USA. Clade YNP-2 contains three subclades (YNP-2-BS1, YNP-2-MV3/5/6, and YNP-2-MV4), the members of which are predominantly from a single site or a set of sites with similar environmental characteristics (see Section 3.6.2). Clade YNP-2 also contains one sequence from mat sample BS3.2 (filled blue star in Figure 7) and all sequences from enrichments from samples BS3.1 and BS3.2 (open blue stars in Figure 7). This suggests that the BS3.1/3.2 site contained at least two genetically distinct *Chlorobaculum* populations, one of which (belonging to clade YNP-2) was favored by our cultural conditions.

##### Correspondence between Clades, Subclades, and Environmental Conditions

Clades and subclades appear to be associated with different environmental factors. For instance, within the YNP-2 clade, subclade YNP-2-MV4 sequences (purple highlight in Figure 7) derive from a hotter acidic site (50.1 °C, pH 5.7), compared to subclade YNP-2-MV3/5/6 sequences (green, yellow, and rust highlight in Figure 7), which derive from cooler acidic sites (34.3 to 40 °C, pH 5.4–5.9). Similarly, sequences of subclade YNP-2-BS1 (dark blue highlight in Figure 7) derive from a near-neutral site with a higher range of temperatures (41.6 to 51.6 °C, pH 6.4–6.8) compared to sequences from clades YNP-1-BS3.1/3.2-MAT (light- and medium-blue highlight in Figure 6), which derive from more alkaline sites with a lower range of temperatures (35.6 to 44 °C, pH 6.4–6.8) and LIP-MAT (solid red circles in Figure 6), which also derive from a lower-temperature site. The association of these clades and subclades with different temperatures and pH ranges suggests that ecological specialization in these parameters may have occurred.

##### Sequence Relatedness and Ecotype Simulation 2 Modeling

The degree of divergence within and between sequences in these clades is presented in Table 3. The YNP-1-BS3.1/3.MAT clade and the LIP-MAT clade sequences both differ from sequences in the TLS, LIP-ENR, and YNP-2 clades by >1.5%. However, the YNP-1-BS3.1/3.2 and LIP-MAT sequences only differ by 0.3 to 0.8%, and the TLS and LIP-ENR sequences only differ by 0.7 to 1.2%. However, ES2 analysis, which predicts ecologically or geographically distinct populations from the sequences sampled based on evolutionary ecology theory, suggests that clades YNP-1-BS3.1/3.2-MAT, LIP-MAT, TLS, and LIP-ENR all represent distinct ecotypes/geotypes (black bars in Figure 6). Subclades in clade YNP-2 differed from each other by 0.2–0.4% to 1.0–1.8%. However, ES 2 analysis predicted that all of the members of subclade YNP-2-BS1 and most of the members of subclades YNP-2-MV4 and YNP-2-MV3/5/6 represent different putative ecotypes (black bars in Figure 7).

##### Enrichment Bias

Figure 7 shows that subclades YNP-2-BS1 and YNP-2-MV4 of clade YNP-2 contained sequences from both mats and enrichments from a single hot spring, and this is also true of the TLS clade (see Figure 6). In contrast, sequences from enrichments derived from the Lipayo-5 sample were not representative of the mat sequences from this site (LIP-MAT sequences, Figure 6), but instead formed a second clade (LIP-ENR sequences, Figure 6), close to the TLS clade. Similar results were also observed for “Big Spring” samples BS3.1 and BS3.2, in which mat sequences formed a distinct clade near the base of the tree (YNP-1-BS3.1/3.2-MAT clade, Figure 6), but one mat sequence and all enrichment sequences were associated with the more peripheral YNP-2 clade (filled and unfilled blue stars in Figure 7, respectively). The populations contributing these sequences must have also been present in the mat samples containing LIP-MAT sequences YNP-1-BS3.1/3.2-MAT sequences, as the enrichments were derived from the mat samples.

## 4. Discussion

To date, the only reports of thermophilic GSB have been those describing mats that were found in hot springs in Rotorua, New Zealand [19] or isolates that were cultivated therefrom [20], and a possible observation based on microscopy alone [21,22] decades ago. It was proposed that high-sulfide habitats with a pH less than 6.5 were required to eliminate other phototrophic bacteria and that this restricted the geographic distributions of these thermophilic GSB [21,22]. Multiple lines of evidence presented here confirm the presence of GSB in a Lipayo hot spring, PHL, and in thermal springs in two distinct regions of YNP, the Mud Volcano region and the Gibbon Geyser Basin, all of which showed pH values of <6.8, but only moderate sulfide concentrations in the 7–42 µM range (Table 1). First, anaerobic, rod-shaped, nonmotile unicellular bacteria that have the ability to photooxidize sulfide and/or thiosulfate were successfully enriched from springs in these regions. Second, the enriched cells contained BChl *c*, chlorobactene, and chlorosomes (Figure 4 and Figure 5), all characteristic features of GSB [17]. Third, phylogenetic analysis of partial 16S rRNA sequences retrieved from mats and enrichments demonstrated that these organisms are most closely related to, but distinct from, *Cba. tepidum* and other cultivated *Chlorobaculum* species. Like *Cba. tepidum*, these GSB were found in habitats up to 54 °C, and enrichment cultures could grow at temperatures up to 44 °C.

### 4.1. Putative Novel *Chlorobaculum* Species

The GSB 16S rRNA sequences from hot spring environments discovered in this study are much more diverse than was heretofore known from the cultivation of *Cba. tepidum* strains TLS^T^ and S.STP.1 from Travel Lodge and Sulfur Point 1 springs in NZ, respectively. Analysis of diversity in a mat sample from TLS showed only variants that were very closely related to the type strain isolate (Figure 6), suggesting that all of these sequences represent organisms affiliated with the established species *Cba. tepidum* [13,20]. In contrast, sequences from hot spring sites in PHL (Lipayo-5) and the USA (YNP) expanded the phylogenetic breadth of hot spring-associated GSB and also the geographic and ecological diversity of these organisms.

Identity values based on 16S rRNA sequences of <98.7% to the nearest type strain sequences are accepted to correlate with genome ANI values below 95%, the currently accepted threshold for species delineation [65]. On the basis of 16S rRNA sequence divergence alone (see Table 3), members of clades YNP-1-BS3.1/3/2-MAT combined with LIP-MAT, members of clades TLS combined with LIP-ENR, and at least some members of clade YNP-2 would be considered to belong to three different species. In the case of clade YNP-2, further evidence comes from the genome sequence of strain MV4-Y, which exhibits ANI values of 88.38% and 85.1% with the genomes of *Cba. tepdium* and *Cba. limnaeum*, respectively. However, partial 16S rRNA sequences of clades YNP-1-BS3.1/3.2-MAT and Lip-MAT obtained in this study differ by only 0.8–1.0%, and members of clades LIP-ENR and TLS differ from each other by only 0.7–1.2% in the partial 16S rRNA gene sequence. Therefore, these four clades would be seen as two distinct species on this basis alone (Table 3). However, at this level of 16S rRNA gene sequence identity, respective ANI values of <95% cannot be readily assumed, nor can one predict the unification of isolates with sequence identity values >99% to represent the same species. It has been shown for many genera that closely related isolates with 16S rRNA gene sequence identities of >99% can belong to different species [66,67]. Additionally, it has been shown that strains of bacteria can be cultivated from a single aquatic sample which have identical 16S rRNA sequences but different 16S-23S rRNA internal transcribed spacer region sequences, are ecologically distinct in their substrate utilization patterns and niches [68]. With these observations in mind, the strong bootstrap support, monophyly, and divergence based on adaptation and geographic isolation inferred from distribution patterns lead us to hypothesize that these clades represent four rather than only two distinct species. Furthermore, theory-based ES2 analysis predicted that all of these clades constitute distinct ecotypes and/or geotypes. Verification of whether this really makes them distinct species would require more information. For instance, while it is clear that YNP-1-BS3.1/3.2-MAT and LIP-MAT clade members are different geotypes, it is not clear that they differ ecologically and would compete with each other if they were in the same place. Technically, if representative isolates and their genomes were available, it might be shown that their ANI values differ enough to be considered distinct species. The same applies for the members of the TLS and LIP-ENR clades.

Members of subclades of clade YNP-2 are too closely related to meet the 16S rRNA criterion separating species mentioned above. However, ES2 analysis predicts that these subclades represent distinct putative ecotypes, and this is consistent with the distribution of their members in springs of different temperatures and/or pHs. Whether or not these constitute different ecological species depends on whether they would compete with each other in these different environments.

In hot spring environments, the presence of “ecological species”, so-called “ecotypes”, was previously shown for mat-inhabiting cyanobacteria in YNP. *Synechococcus* populations with closely related or even identical partial 16S rRNA sequences differed in internal transcribed spacer region sequences [52] and PsaA gene sequences [53,54]), markers that offer greater molecular resolution. Putative ecotypes predicted from these sequences by ES analysis showed different spatial (horizontal and vertical) distributions in the mats [53], indicating specific adaptations to physical parameters and differing conditions, such as light and temperature. Specific adaptations to physical parameters and conditions in the mats, such as to light [69] and temperature [70], were confirmed by studying representative isolates. The genomes of isolates with different adaptations to light exhibited ANI values of 98.5 to 99.05%, and low-light-adapted strains contained a gene cassette encoding a novel allophycocyanin that extends the light spectrum absorbed, which strains adapted to high light did not contain [71]. Similar adaptations might be found for the distinct *Chlorobaculum* populations represented by the different sequences observed and putative ecotypes indicated in this study. Although genome analyses of the distinct *Synechococcus* spp. isolates did not indicate them to be different species in the currently proposed understanding of showing <95% ANI values [71], based on their specific adaptations, these strains behave differently, are found in different niches, and outcompete each other under certain conditions [53], and would therefore be considered different ‘ecological species’.

### 4.2. Are the New GSB Species Thermophilic?

Wahlund et al. [20] described *Cba. tepidum* as a thermophile on the grounds that it was cultivated from moderately thermal springs at 45–55 °C and could grow between 32 and 51–52 °C (optimally at 47–48 °C). A second anoxygenic phototrophic bacterium isolated from the same sample, *Allochromatium tepidum* strain NZ, also turned out to show a thermophilic character with a temperature growth range of 30–50 °C (optimum 40–45 °C [72]. Although our survey of GSB found in hot spring environments was designed to use temperatures similar to those of NZ springs studied and used to cultivate *Cba. tepidum* therefrom, we needed to vary enrichment temperatures according to where and under what conditions GSB formed mats in PHL and YNP springs and did not conduct an extensive investigation of the temperature relations of enriched strains. Many of our enrichments came from high-temperature environments, with temperatures clearly above the values used for cultivation. and grew well at 44 °C, which is within the range of temperatures and near the optimum temperature for *Cba. tepidum*. In one case, specifically, 16S rRNA gene sequences affiliated with the *Chlorobaculum* sp. YNP-2 clade were obtained from a sample with an in-situ temperature of 67 °C. Due to the consistently high temperatures in their natural environments, we hypothesize that these organisms (especially those observed in YNP) show specific adaptations to higher temperatures and have a thermophilic character. However, growth experiments at different temperatures would be needed to specify the growth range and optimal temperatures to confirm our hypothesis.

### 4.3. Cultivation Bias

Interestingly, 16S rRNA gene sequences from enrichment cultures did not always reflect sequences obtained directly from the same mat sample. This was noted in the case of enrichments from site Lipayo-5, the sequences from which formed a distinct clade (LIP-ENR) near the TLS clade, and from enrichments from “Big Spring” sites BS3.1 and BS3.2, which also contained sequences that were associated with the YNP-2 clade. The growth medium used for enriching GSB was the same Maintenance Medium used for the three NZ *Cba. tepidum* strains and did not necessarily reflect environmental conditions where and when these samples were collected. This could introduce an enrichment bias favoring populations that grow fastest on the enrichment medium but are not necessarily the largest populations present in the mat [73].

One possible explanation for the coexistence of multiple ecotypes in a single sample is that fluctuating environmental conditions could favor different populations at different times. We noted above that the BS3.1/3.2 mat site was located between an acidic inlet and an alkaline hot spring vent and noted that the pool level rose and fell periodically, changing the pH and temperature of the water overlying the mat. The same situation existed at the Lipayo-5 site, where the main stream was hot and acidic, but a side stream entered just above the GSB mat, lowering the temperature and raising the pH. A change in the volume of flow in either of these streams would likely change the temperature and pH of the GSB mat. Another possible explanation is provided in studies of hot spring *Synechococcus* spp., where ecotypes that predominate at higher-temperature upstream sites were shown to be transported to cooler sites downstream, where low-temperature-adapted ecotypes predominate [74]. Overgrowth of more abundant low-temperature-adapted *Synechococcus* spp. by less abundant high-temperature-adapted *Synechococcus* spp., both present in the low-temperature samples, occurred when these samples were shifted to higher temperatures [53,75].

### 4.4. Conditions in Which Hot Spring-Associated GSB May Be Found

It was previously proposed that the development of GSB mats in hot springs might require a special mix of conditions, including high (mM) sulfide concentrations, a pH of less than 6.5, and temperatures between 40 and 55 °C [19,21]. It was presumed that the low pH prevented the growth of *Chloroflexus* and that the combination of high temperature, high sulfide, and low pH prevented the growth of cyanobacteria; in the absence of oxygen and competition, thermophilic GSB thrived. As such conditions are rare in nature, it was further suggested that this might explain why the geographic distribution of thermophilic GSB is limited [19,21]. Our findings necessitate a rethinking of this assessment. Enrichment and/or 16S rRNA evidence of GSB in the Lipayo spring, “Big Spring”, Mud Volcano, and two Mammoth hot springs (Narrow Gauge and Angel Terrace) expand the geographic and ecological ranges of these organisms. First, the sulfide levels of the newly discovered springs containing GSB were much lower than those of the Rotorua area springs. For instance, whereas TLS contained 0.28 to 1.85 mM sulfide, the Lipayo spring and YNP springs we studied contained only 1.3 to 40 µM sulfide. Second, whereas the pH values of the Lipayo hot spring and Mud Volcano sites were similar, ranging from 5.4 to 6.0, the pH values of the YNP “Big Spring” sites and the two Mammoth springs mentioned above were 6.2 to 6.9. Third, while the Lipayo-5 site and most YNP springs had temperatures similar to those of Rotorua springs, a 16S rRNA sequence was readily amplified from a 67.4 °C site in Angel Terrace Spring, Mammoth, YNP, possibly indicating that thermophilic GSB can tolerate higher temperatures than previously thought. However, we were unable to obtain an enrichment from this site at 44, 49, or 55 °C. Therefore, it is possible that although this sequence was present, the organism represented by it may have a lower temperature limit or may not have been growing at 67.4 °C. Fourth, it appears that GSB can coexist with other phototrophic organisms. In most springs, GSB mats occurred as a thin, dense yellowish-green layer above muddy sediments (Figure 2 and Figure 3). Although co-existence with filamentous anoxygenic phototrophic members of the *Chloroflexota* was not observed in this study, at site Lipayo-5, cyanobacteria were detected by 16S rRNA analysis, and in “Big Spring” sites, GSB were detected in samples where cyanobacteria and purple sulfur bacteria were observed through microscopic and pigment analysis (not shown). GSB are considered strict anaerobes [16,17,18], so unless these GSB inhabit anoxic microniches resulting from intense oxygen consumption or can survive in the presence of oxygen, it may be that the cyanobacteria in these mats do not produce oxygen. The ability of cyanobacteria to tolerate sulfide or to switch to anoxygenic photosynthesis in sulfidic environments has long been known [76]. Finally, the existence of and the separate phylogenetic histories of GSB in YNP and PHL hot springs clearly document that the biogeographic distribution of thermophilic *Chlorobaculum* spp. is not limited to NZ hot springs.

## 5. Conclusions

Multiple lines of evidence demonstrate the occurrence of hot spring-associated *Chlorobaculum* spp. not only in TLS and Sulphur Pointe Spring in Rotorua, NZ, but also in the Lipayo-5 site in PHL and two areas of YNP, USA, particularly Mud Volcano and Gibbon Geyser Basin, with the possibility that such organisms may also occur in springs in the Mammoth Hot Springs area as well. Based on the degree of 16S rRNA divergence alone, one might hypothesize that there exist at least three *Chlorobaculum* species, represented by (i) clade LIP-MAT together with clade YNP-1-BS3.1/3.2-MAT, (ii) clade TLS together with clade LIP-ENR, and (iii) *Chlorobaculum* strain MV4, a member of clade YNP-2. However, considering monophyly, tight 16S rRNA sequence clustering, geographic and ecological distribution patterns, evolutionary ecological simulation, and the fact that even organisms within a single sample that have identical 16S rRNA sequences can be ecologically and taxonomically distinct, we hypothesize that members of clades YNP-1-BS3.1/3.2-MAT, LIP-MAT, TLS, LIP-ENR, and *Chlorobaculum* strain MV4-Y in clade YNP-2 all represent distinct *Chlorobaculum* species. Using these criteria, we further hypothesize that, within clade YNP-2, the subclade of clade YNP-2 that contains the MV4 sequence and the other two major subclades could each be considered ecological species. The phenotypic properties of these species inferred from distribution patterns in springs of different temperatures and pHs suggest the high likelihood that moderately thermophilic GSB species are present in PHL and YNP. However, cultivation and further characterization will be required to prove that these are indeed new species with the inferred adaptations.

The separate clades of 16S rRNA gene sequences for variants in NZ, the PHL and YNP, suggest that there has been divergence due to geographic isolation, as has been found for hot spring cyanobacteria [54,77] and *Chloroflexota* [78]. The observance of two YNP clades, one containing only members from lower-temperature near-neutral pH sites at “Big Spring” and the other containing very closely related subclades with members obtained from sites of different temperatures and pH values, suggests that ecological divergence has also occurred in these organisms, as has been found for hot spring cyanobacteria [53]. Although the sequences in many clade YNP-2 subclades differ by only a few nucleotides, such small-scale differences in the 16S rRNA sequence were shown to be ecologically significant in hot-spring *Synechococcus* spp. [53]. The geographical and ecological patterning of the distribution of variation in these very different taxa speaks to the generality of the forces that drive diversification, at least among hot spring microorganisms. This newly discovered diversity of hot spring-associated GSB, which grow rapidly and densely at 40–44 °C, may offer new research tools for investigating thermophilic GSB, and the observation of possible GSB in environments as hot as 67 °C will hopefully spur continued interest in the possible existence of extremely thermophilic GSB and thereby aid in a better understanding of the early evolution of photosynthesis.

## Figures and Tables

**Figure 1 microorganisms-11-02921-f001:**
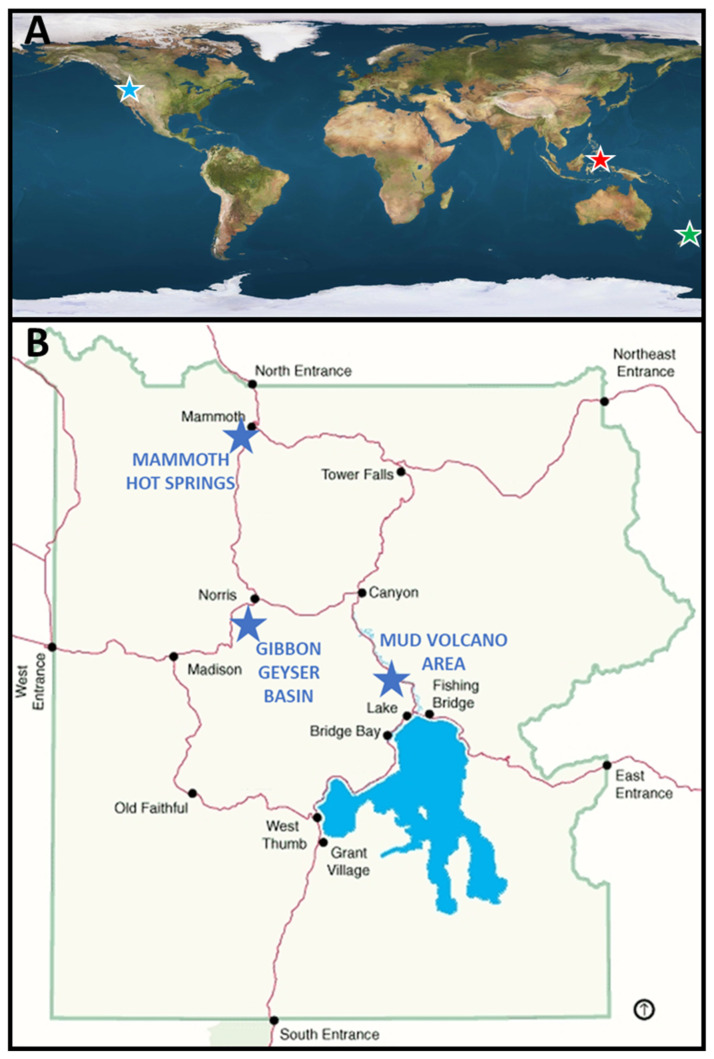
Locations sampled are (**A**) New Zealand (green star) and the Philippines (red star), and (**B**) Yellowstone National Park, USA (blue stars).

**Figure 2 microorganisms-11-02921-f002:**
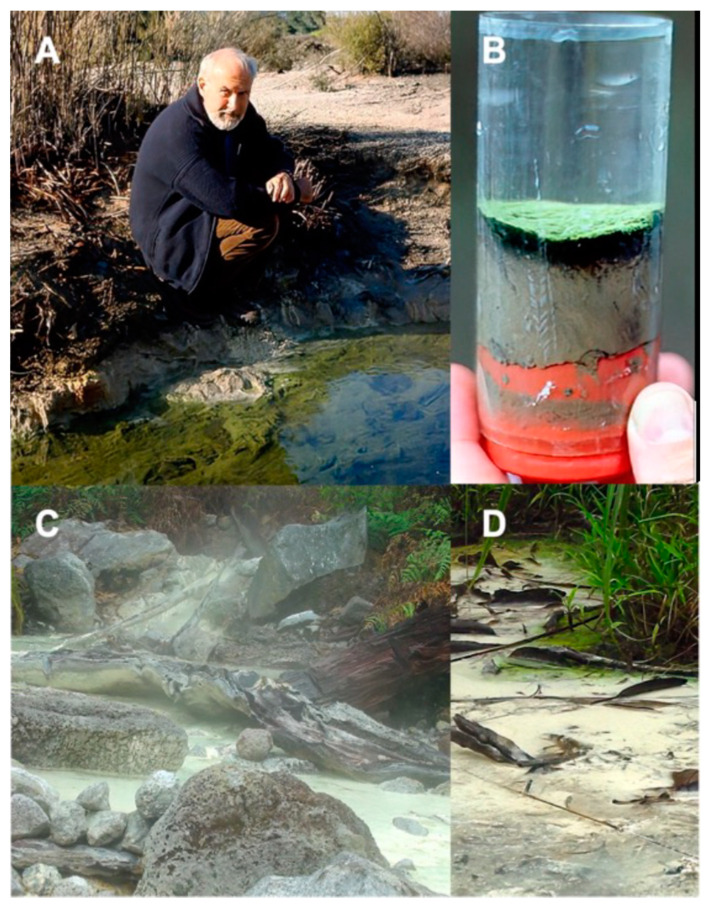
Sites where mats containing green sulfur bacteria were found were in New Zealand and the Philippines. (**A**) Dick Castenholz at Travel Lodge Stream, Rotorua, NZ, in 1986. (**B**) Core of the Travel Lodge Spring mat. (**C**) Lipayo hot spring, the Philippines. (**D**) Thin yellowish-green veil of GSB near the shore at site Lipayo 5.

**Figure 3 microorganisms-11-02921-f003:**
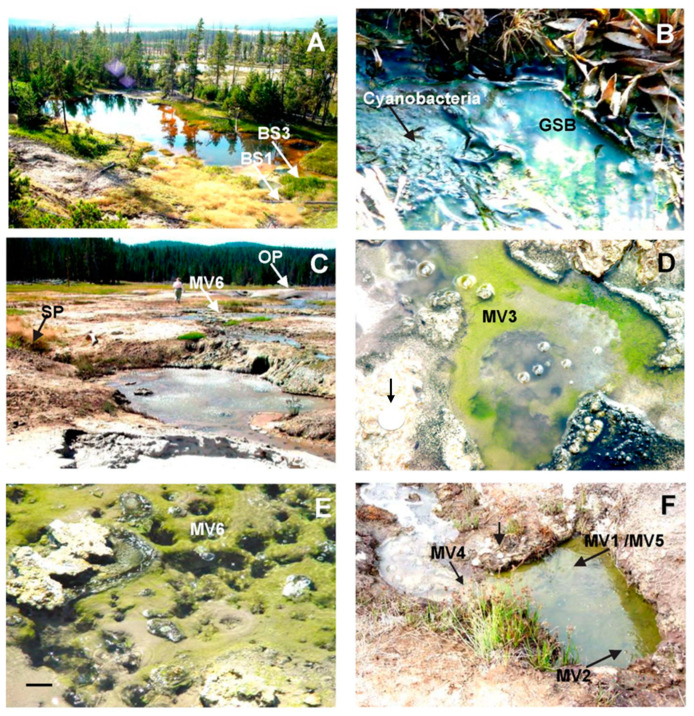
Sites where mats containing green sulfur bacteria were found in Yellowstone National Park. (**A**) “Big Spring.” The arrows show the collection points where GSB mats were found in this pool. (**B**) Comparison of the GSB mat and a cyanobacterial mat at site BS1. (**C**) The effluent stream originating from Obsidian Pool (OP) in the Mud Volcano region. Sites where mats containing GSB were found are indicated by the arrows. SP, side pool, where sites MV1 and MV2 were located in 2000 and sites MV4 and MV5 were located in 2001 (see panel F). (**D**) Site MV3 bubbling spring and GSB mat. Note the 25 mm coin for scale (unlabeled black arrow). (**E**) Pool at the source of site MV6 with numerous bubbling springs with mats containing GSB. Scale bar (25 mm) (**F**) Shallow side pool (SP) and sediment covered with a GSB mat in 2001. Source of samples MV4 and MV5; the same pool was the source of samples MV1 and MV2 in 2000, when they were discrete mats. Note the 25 mm coin for scale (unlabeled black arrow).

**Figure 4 microorganisms-11-02921-f004:**
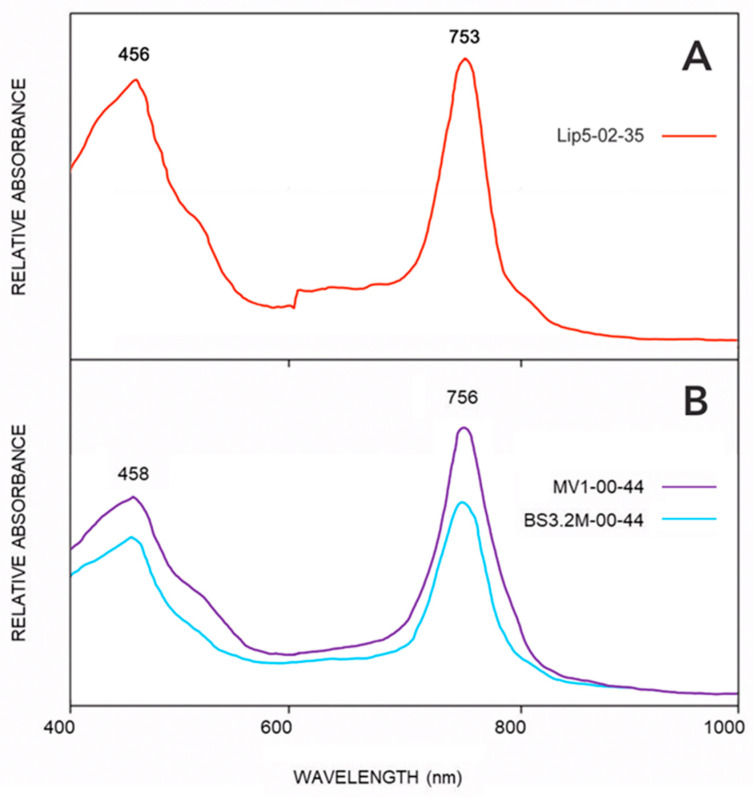
Whole-cell absorbance spectra from GSB enrichments. (**A**) From Lipayo 5, grown at 35 °C. (**B**) From sites MV1 and BS3.2M, grown at 44 °C. The absorbance peaks at 753 nm and 756 nm are due to BChl *c* aggregates in chlorosomes. Absorbance peaks at 456–458 are due to chlorobactene. Spectra for all enrichments were taken and showed the characteristic peaks for BCl *c* and chlorobactene.

**Figure 5 microorganisms-11-02921-f005:**
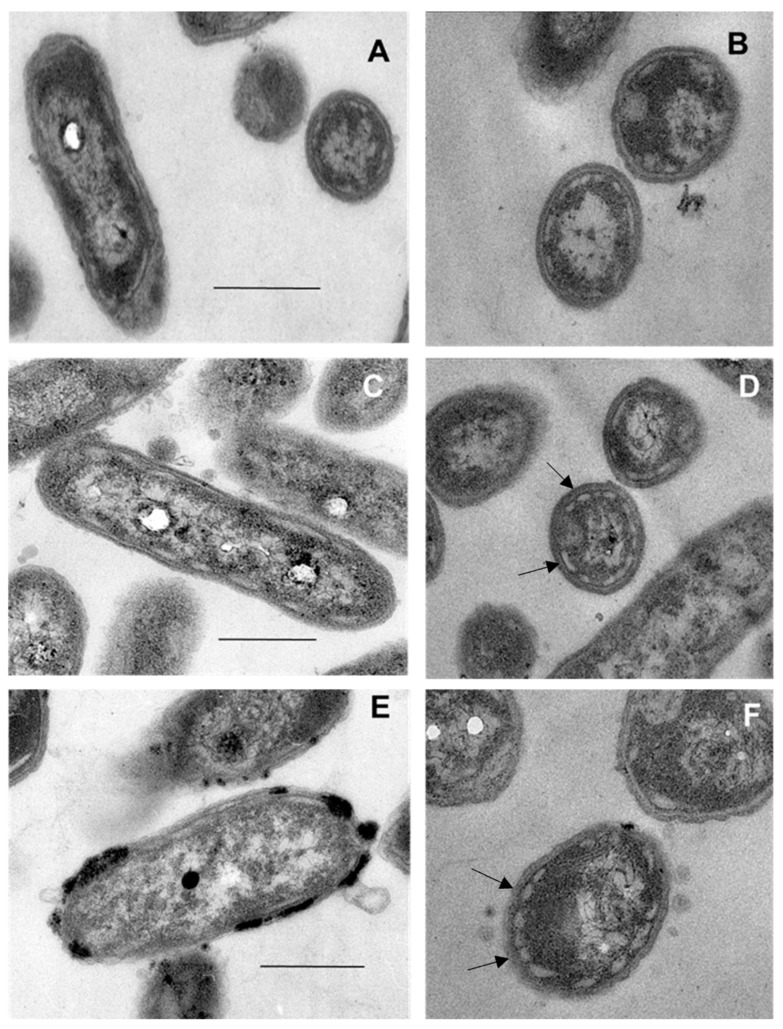
Thin-section electron micrographs show chlorosomes in cells from GSB enrichments from YNP hot springs. (**A**,**B**) cells from site BS1 enrichment. (**C**,**D**) cells from site BS3.2M enrichment. (**E**,**F**) cells from site MV4 enrichment. Arrows show examples of well-resolved chlorosomes, which are the more lightly-stained, irregular, sac-like objects closely appressed to the inner surface of the cytoplasmic membranes. The magnification for all micrographs was 80,000×. The size bars represent 250 nm.

**Figure 6 microorganisms-11-02921-f006:**
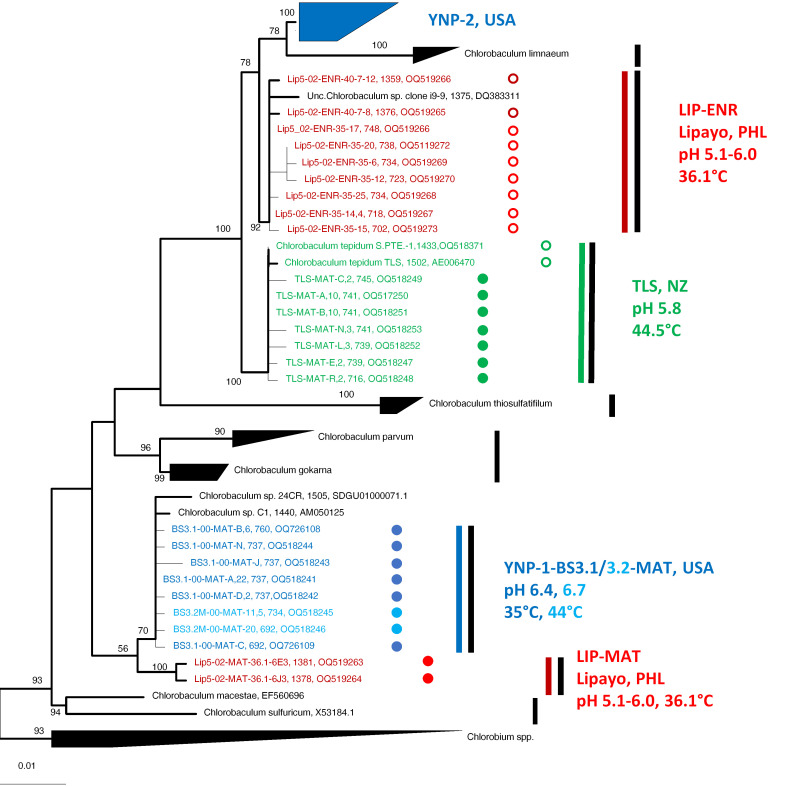
A phylogenetic tree of 16S rRNA gene sequences shows relationships among hot spring variants and cultivated *Chlorobaculum* strains. The tree was rooted by sequences of several *Chlorobium* strains. Sequence labels include site, year collected (00, 2000; 02, 2002), whether from mat (MAT) or enrichment (ENR), temperature grown at (enrichments only), sequence designation, number of copies of the sequence (if more than 1), sequence length (nt), and accession number. Sequences and circles indicating association with mat (closed) or enrichment (open) are color-highlighted based on the samples from which they were derived. Colored vertical bars and associated text show relationships of clades to mat and enrichment samples from different geographic regions and environmental conditions. Black vertical bars represent putative ecotypes/geotypes demarcated by Ecotype Simulation 2 analysis. Numbers at nodes are bootstrap values from 100 replicates. The scale bar indicates the number of substitutions per nt position.

**Figure 7 microorganisms-11-02921-f007:**
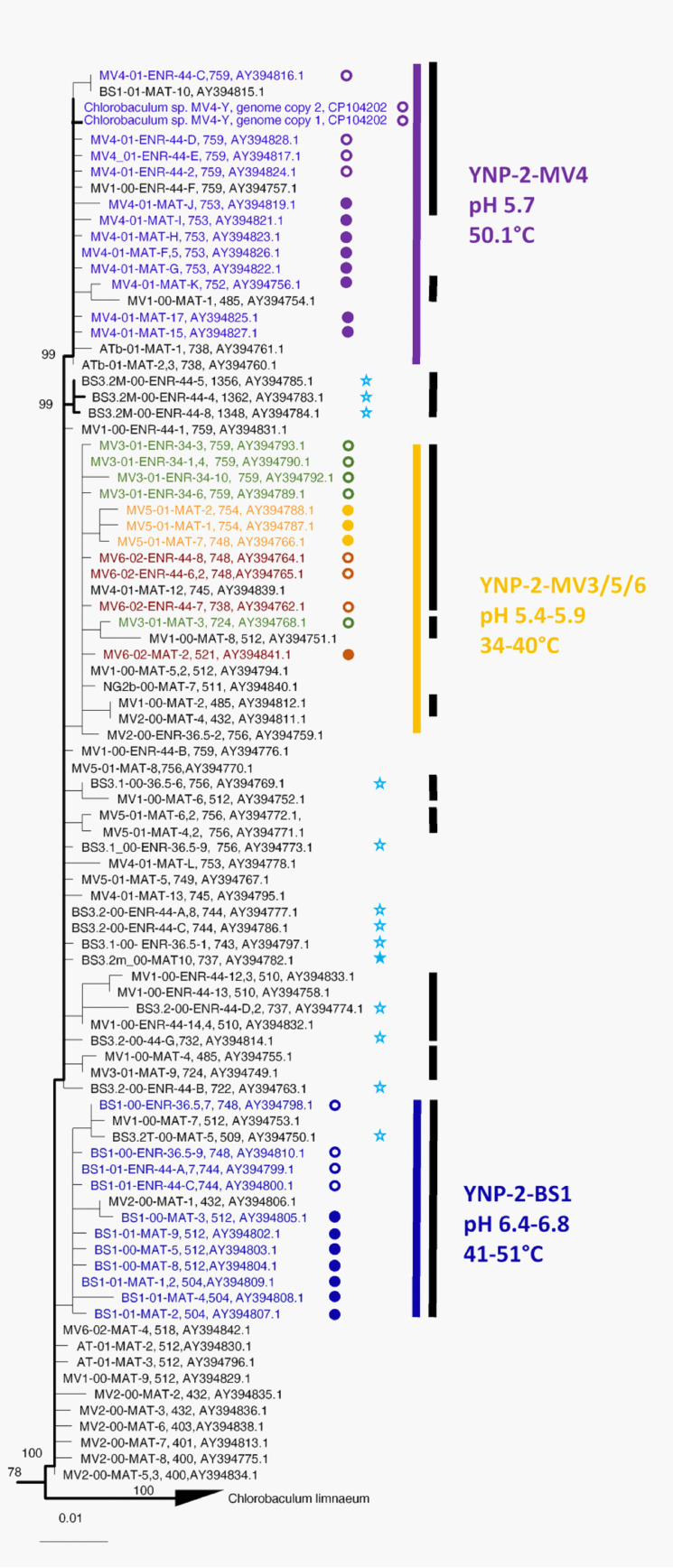
The expanded phylogeny of the YNP-2 clade is shown in Figure 6. Sequence labels include site, year collected (00, 2000; 02, 2002), whether from mat (MAT, closed circles) or enrichment (ENR, open circles), temperature grown at (enrichments only), sequence designation, number of copies of the sequence (if more than 1), sequence length (nt), and accession number. Sequences and circles indicating association with mat (closed) or enrichment (open) are color-highlighted based on the samples from which they were derived. Blue stars highlight sequences that came from the site YNP-1-BS3.1/3.2 mat (filled) and enrichment (unfilled) samples. Colored vertical bars and associated text show relationships of clades to mat and enrichment samples from springs with different environmental conditions. Black vertical bars represent putative ecotypes/geotypes demarcated by Ecotype Simulation 2 analysis. Numbers at nodes are bootstrap values from 100 replicates. The scale bar indicates the number of substitutions per nt position.

**Table 1 microorganisms-11-02921-t001:** Hot springs in which green sulfur bacteria were found in New Zealand (NZ, green), The Philippines (PHL, red) and YNP (blue).

Site	Sample Date	Site Description	Temp °C	pH	Sulfide µM	Mat DNA GSB PCR product				GSB Enrichment		
						600F	Nested 600F	619F	27F	Temp °C	Gr ^a^	PCR Prod
**NZ**												
TLS	9/9/1986	Ropey green film atop mud	44.5	5.8	280–1850	+	NT	NT	NT	NT	NT	NT
**PHL**												
Lip5	22/1/2002	Grass green growth on the surface at the edge of the stream. See Figure 2D	36.1	5.1–6.0	40	NT	NT	NT	+	36.5 ^b^	+	+
**YNP**												
BS1-00 ^c^	21/6/2000	Bright green mat in a muddy area	41.6	6.4	NT	No	No	+	NT	36.5	+	+
BS1-01 ^c^	6/8/2001	Bright green mat in muddy area	51.6	6.8	1.8	No	No	+	NT	44	+	+
BS2	21/6/2000	Yellowish-green mat on mud with thin white film (sulfur?) on top	51.7 ^d^–46	4.6 ^d^–6.4	NT	No	No	+	NT	50	No	NT
BS3.1	21/6/2000	Pinkish-purple/yellowish green mat, floating on surface	35.6	6.4	NT	+	NT	NT	NT	36.5	+	+
BS3.2T ^e^	29/6/2000	As BS3.1, but larger	44.4	6.7	NT	No	+	NT	NT	44	+	+
BS3.2M ^e^	29/6/2000	Deep green layer, and below that, black mud and white filaments	44.4	6.7	NT	No	+	NT	NT	44	+	+
MV1	29/6/2000	Bright green mat circling a small bubbling spring in the mud	49–54	5.9	NT	No	No	+	NT	44	+	+
MV2	29/6/2000	Bright green mat circling small bubbling spring on mud	36–44	5.7	NT	No	+	+	NT	36.5	+	+
MV3	7/8/2001	Yellowish-green mat on mud around bubbling spring. See Figure 3D	34.3	5.8	2.7	No	+	NT	NT	34	+	+
MV4	7/8/2001	Yellowish-green mat on mud	50.1	5.7	5.3	+	+	NT	NT	44	+	+
MV5	7/8/2001	Yellowish-green mat on mud	39–40	5.7–5.9	4.7	+	+	+	NT	38	+	+
MV6	31/7/2002	Yellowish-green mat on mud circling a bubbling spring. See Figure 3E	37–40	5.4–5.7	1.3	+	+	NT	NT	44 ^f^	+	+
AT	11/8/2001	Olive green to orange mat sample in a shallow travertine pool near its top	67.4	6.9	27.7	+	NT	+	NT	55 ^g^	No	No
NG2	14/6/2000	Fibrous dark olive green and orange mat in a tiny spring near the midpoint of Narrow Gauge	52.5	6.2	NT	No	NT	+	NT	52	No	No

NT = Not Tested; ^a^ Gr = growth; ^b^ It also grew at 40 °C, though not as well. ^c^ The last two digits indicate the year sample. ^d^ The water depth was rising and falling in minutes, resulting in fluctuations in temperature and pH. ^e^ These two samples were from the middle (M) and top (T) layers of the mat. ^f^ The inoculum for this enrichment was a 5 × 10^−7^ dilution. ^g^ Also tested at 44 °C and 49 °C.

**Table 2 microorganisms-11-02921-t002:** PCR primers used in this study.

Primer	Specificity	Sequence 5′ to 3′	Reference
27F	EUB	AGA GTT TGA TCC TGG CTC AG	[36,37]
1492R	UNIV	GGT TAC CTT GTT ACG ACT T	[36,37]
GS.619F	GSB	GGG GTT AAA TCC ATG TGT GCT	[35]
GS.1144R	GSB	CAG TTC ART TAG AGT CC	[35]
GSB600F	GSB	GGG GGT TAA ATC CAT GTG	This study
1392R	UNIV	ACG GGC GGT GTG TAC	[38]

**Table 3 microorganisms-11-02921-t003:** Percent divergence in 16S rRNA sequence within (shaded) and among (unshaded) members of clades and subclades.

Species/Clade	*Cba. parvum*	*Cba. thiosulfatiphilum*	*Cba. tepidum*	*Cba. limnaeum*	LIP-MAT	YNP-1-BS3.1/3.2-MAT	TLS	LIP-ENR	YNP-2	YNP-2-MV4	YNP-2-MV3/4/5	YNP-2-BS1
*Cba. parvum* (7)	0–1.7											
*Cba. thiosulfatiphilum* (2)	3.4–4.9	0.4										
*Cba. tepidum* (2)	2.8–4.4	3.5–3.9	0									
*Cba. limnaeum* (4)	3.2–4,9	3.3–3.9	1.9–2.5	0–0.6								
LIP-MAT	3.1–4.0	4.0–4.4	3.6–3.8	1.2–2.2	0.0–0.2							
YNP-1-BS3.1/3.2-MAT	2.6–4.7	3.5–4.1	2.9–3.5	1.7–2.6	0.3–0.8	0.2						
TLS	2.2–4.1	3.5–4.1	0.0–0.4	1.5–2.5	2.7–3.1	2.1–3.1	0.1–0.6					
LIP-ENR	1.7–4.1	3.3–4.1	0.8–1.3	1.2–2.2	1.7––3.6	1.9–3.4	0.7–1.2	0.0–0.4				
YNP-2	1.5–5.5	2.0–5.3	1.4–2.8	1.2–3.0	2.2–4.1	2.7–4.5	1.2–2.5	0.8–2.1	0–2.0			
YNP-2 MV4	2.9–4.9	2.9–5.0	1.5–2.0	1.3–2.2	2.9–3.6	2.8–4.0	1.6–2.2	0.9–1.8		0.0–1.0		
YNP-2-MV3/4/5	1.9–5.0	2.3–5.0	1.6–2.1	1.3–2.2	2.5–3.7	2.7–4.5	1.4–2.5	1.0–2.0		0.2–1.7	0.0–1.2	
YNP-2-BS1	1.6–5.3	2.6–3.6	1.4–2.0	1.7–2.6	2.3–3.4	2.7–4.2	2.3–3.4	0.8–1.9		0.2–1.2	0.4–1.8	0.0–1.0

## Data Availability

The datasets in this study can be found in online repositories. The names of the repositories and accession numbers can be found below: 16S rRNA sequence data, accession: PopSet: 40556720 and accession numbers OQ518241-OQ518253, OQ518371, Q519263-OQ519273, OQ726108-OQ726109, and OQ913398-OQ913416.

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
