# Peer review of "Geographic and Ecological Diversity of Green Sulfur Bacteria in Hot Spring Mat Communities"

_microorganisms, 2023, doi:10.3390/microorganisms11122921_

Round 1

Reviewer 1 Report

Comments and Suggestions for Authors

Green sulfur bacteria (GSB) form an independent phylum Chlorobiota. Representatives of GSB have a number of features that distinguish them from other anoxygenic phototrophic bacteria. There is reason to believe that they played an important geochemical role in the early stages of the evolution of planet Earth. It remains a mystery why there are no extremely thermophilic representatives among the GSB? At the time of this work, only one moderately thermophilic species, Chlorobaculum tepidum, was known to exist. This circumstance determines the relevance of the tasks posed in the work under review.

As a result of the work done, it was proven that moderately thermophilic GSBs are widespread in mesothermal springs in three geographically distant areas. These springs had a slightly acidic to near neutral pH and contained low concentrations of sulfide. It was established that the localization and individual hydrochemical characteristics of the sources were reflected in the genetic properties of the discovered natural GSB phylotypes and isolated cultures. It has been proven that ecological divergence took place among moderately thermophilic GSB. 16S rRNA gene sequences from microbial mats collected in hydrothermal waters of New Zealand, Thailand and Yellowstone (USA) formed separate phylogenetic clades, which indicates the geographical isolation of thermophilic GSB clones. Note that in the study positions of ecotypes, a simulation model of evolutionary ecology was used, based on a model of a stable ecotype of species, and predicting speciation.

The advantage of the work is that the authors were able to use samples of mats frozen in different years, including isolating from these several cultures of bacteria related to Chlorobaculum tepidum. The authors note that 16S rRNA gene sequences from enrichment cultures do not always reflect sequences obtained directly from the same mat sample. This fact once again proves that the isolation and study of microbial cultures significantly complements the data obtained from genetic analysis of natural microbial communities.

The authors suggest that in the studied hydrotherms, in addition to Chlorobaculum tepidum, there are several new species of moderately thermophilic bacteria. Of course, obtaining cultures and their additional research will be required for a valid description of new taxa.

In conclusion, I note that the scientific level of the work under review is quite worthy for publication in the journal Microorganisms without significant changes.

As a remark, I note that, in my opinion, when discussing the obtained material, there is not enough discussion about the evolution of representatives (members) of Chlorobiota from moderate thermophiles to mesophylls, dominant in modern meromictic water bodies and in low-temperature microbial mats. The question also arises: why are there no extremely thermophilic Chlorobiota, in contrast to the phototrophic Chloroflexota?

Author Response

  1. Why are there no extremely thermophilic GSB?:
    1. One possibility is that we have failed to find them. We did observe GSB 16S rRNA sequences in the 67.4C Mammoth hot spring AT mat samples, but failed to grow them in enrichments performed at 44-55C. Reviewer 2 had concern about why enrichments from these samples were done at a temperature well below that of the habitat. It is possible that we might have cultivated an extremely thermophilic GSB had we incubated at a temperature closer to the habitat temperature, presuming that extremely thermophilic GSB have a lower-temperature limit above the enrichment temperatures we used. We revised the last sentence of the Conclusions section to point this out. It is our hope that this study and the discovery of GSB DNA sequences in a mat at 67.4C will encourage other researchers to explore more hot springs for GSB at higher temperatures.
    2. If extremely thermophilic GSB really don’t exist, we can only speculate that there must be some inherent sensitivity of their unique photosystem to temperatures above ~55C.
  2. Not enough discussion about the evolution of representatives of Chlorobiota from moderate thermophiles to mesophylls:
    1. Although it is tempting to address this by walking through the phylogenies shown, we decided to follow the advice of Reviewer 2 and added a Discussion section explaining why inference of thermophyly without traditional growth-temperature responses is risky. Thus, we do not feel that it is appropriate to speculate on evolutionary events at present.
    2. However, we hope that our discovery of putative novel Chlorobaculum species that grow rapidly and densely at 40 to 45C will provide research tools for future investigators and enable more insight into the evolution of photosynthesis.

Reviewer 2 Report

Comments and Suggestions for Authors

This manuscript provides useful information from field site research looking at the diversity of GSB in three, separate, unique environments: New Zealand, Yellowstone National Park, and the Philippines. However, I disagree with the authors’ claims about thermophilic GSB being present in YNP and PHL. I think the manuscript would be more accurate if the authors discussed the enrichments as mesophilic, especially given some of the enrichnment incubation temperatures (i.e., 37 C), with the potential to contain unique thermophilic species, but that further experiments are needed to verify the claim that thermophilic GSB are present in YNP and PHL.

In addition, because of the long period of time between when the environmental samples were taken, when the enrichments were performed, and when the analyses were done, it would serve the authors to be as clear as possible about when sampling/enrichments/analyses took place.

Lastly, again regarding the thermophilic claims, I believe that the title is completely appropriate, but is derailed by the first sentences of the abstract focusing on thermophilic GSB (i.e., considering GSB found in YNP are most closely related to C. limnaeum, despite having been enriched at a higher temperature). It is hard to make firm conclusions without isolates and with the sporadic evidence provided here for different environmental samples. I suggest that the authors provide more supplementary material and revise the text to more accurately reflect the data that mesophilic GSB are present in YNP and PHL, but that the potential for the presence of thermophilic GSB is high.

Line 87: Figure 1 is not particularly insightful and would probably fit better in supplementary material.

Line 124-125: This sentence is saying that enrichments occurred in 1986 or 2000-2002 for all samples?

Table 1 – Site descriptions for MV3 and MV6 say to see Figure 1D and 1E, respectively, but Figure 1 does not have a D or E. It may also be helpful to include in this table where each sample was from (Phillippines, New Zealand, Yellowstone) – since three different hot spring systems were assessed.

Section 2.3 – Microscopy is described as being completed, but no data are included – include in supplementary if necessary.

Section 2.4

Lines 145-149: The authors state that “best” results were obtained when media also contained 4 mM sodium thiosulfate. So were multiple enrichments performed under various conditions (i.e., temperature and media formulations)? It would suit the authors to provide the exact medium components here, so data can be replicated or methods can be used in future enrichments for GSB, if desired.

Line 152, 154-155: why were the specific enrichment temperatures chosen? Why were the enrichment temperatures not matched to the actual environmental temperature at the site?

Lines 156-159: why were the mat/site temperatures not used? (i.e., it can’t be considered surprising that the enrichment for AT “failed” when the enrichment temperature was 12 degrees cooler.

Section 2.5

Line162-163: Does “adjusted to 10% DMSO” mean that DMSO was added to culture aliquots to achieve a final concentration of 10% DMSO before freezing?

Line 163: What was the revival medium?

Line 168: It takes the reader a minute to realize that “enrichment MV4-44” is the revived cultures from site MV4 at 44 C. You might put “(MV4-44)” at the end of the first sentence of this section to inform the reader that you are using this naming scheme.

Line 170: Mention that the pure culture was named MV4-Y. Are there methods/conditions for DNA extraction and sequencing of this strain?

Section 2.6 – are more specific information known? For example, speed of centrifuge, length of time of centrifuge?

Sections 2.6, 2.7, 2.8: were these analyses and extractions performed on the enrichments in 1986/2000-2002 after being grown for three weeks, or were frozen samples revived and these analyses were performed?

Lines 221-224: why might the GSB600F/1392R primer set not yield any GSB sequences?

Section 2.12 – was phylogenetic analysis performed in 2000-2002 or recently?

Figure 3 – it may be helpful to label or have an arrow pointing to the coin in Figure 3D, as in 3F.

Sections 3.1.1 to 3.1.3 – much of this information might be more useful in the methods section.

Line 367-368: what was the timeline of the rising and falling of the water level? Seconds, minutes, hours?

Lines 384-392: was the phase-contrast microscopy performed in 2000 when the samples were collected? Are there any pictures? Include them in the supplemental.

Line 432: Are there images for the fluorescence microscopy for this sample? Include in supplementary data.

Lines 444-446: This sentence seems to indicate that enrichments were revived after 2-10 years, and adds to the confusion of when all of these different analyses took place. What does “remain stable” mean in this context?

Lines 451-453: Additional spectra from all enrichments can be included in supplementary material.

Figure 5 – it may be helpful to point out the chlorosomes in the figure.

Section 3.5 – grown at 44 C? Some of this information can go in the methods section.

Lines 488-489: It should be clarified in the methods that this strain was sequenced elsewhere and that only the genome comparison via ANI was completed here.

Section 3.6.1 – the formatting is messed up here for the title of this section on line 494. Were these PCR products obtained recently or when the enrichments occurred in 1986/2000-2002?

Figure 6 – there is a lost green open circle near the 44.5 C label and a partial extra degrees C symbol in red to the right of Lip-Mat, PHL. I am also not sure if the colored vertical bars are necessary. Are they providing any unique information that the open/closed circles do not already provide?

Table 3 – why are certain cells highlighted in red?

Section 4- Discussion

Lines 10-14: These sentences, following from the previous sentence starting with “Multiple lines of evidence presented here confirm the presence…”, indicate that part of the work presented here (but data not shown or mentioned in results) includes further characterization and analyses of the enrichments (performed in 2000-2002?) such as data related to photooxidation of sulfide/thiosulfate. These data should be included and discussed if it is going to be used as evidence. The sentence at lines 12-13 also states that the “enriched cells contained BChl c, chlorobactene, and chlorosomes…” The authors should reference Figure 4 (for BChl c and chlorobactene) and Figure 5 for chlorosomes. Also, the caption for Figure 4 should remind the reader that the peaks at 456-458 nm are representative of chlorobactene. Also, the evidence provided for these (BChl c, chlorobactene, and chlorosomes) are only present together in samples from site BS3.2M. For example, chlorosomes from BS1 and MV5 sites are shown in Figure 5, but absorption spectra for these samples is not shown. Were spectra taken? If so, did they show spectral features representative of GSB?

Lines 14-18: Much of the author’s claims revolve around the discovery of thermophilic GSB found elsewhere aside from New Zealand. However, the data provided do not support that conclusion – from Figure 6, only the TLS, NZ samples cluster with C. tepidum. The samples from PHL and YNP are all more closely related to non-thermophilic species of GSB. The only evidence for thermophilic ability appears to be the temperature of the actual environmental site; whereas the enrichment temperature for Lip5 and BS1-00 was 36.5 Celsius (not considered thermophilic), and the remaining enrichment temperatures for the majority of sites discussed is 44 C (which is at the very low end of the definition of thermophilic, depending on the reference). I think it would serve the authors to revise the text in a number of places where it is implied that thermophilic GSB have been isolated from sites other than TLS, NZ. A good example of this is in lines 25-28, but should be more consistent throughout the text.

Lines 77-78: would not single out thermophilic Chlorobaculum here since the authors are likely also referring to mesophilic species in their samples.

Lines 79-83: Is this sentence in reference to the previous studies of Synechococcus or are the authors comparing the enrichments in this study to the previous study of Synechococcus? If it is a comparison, I would recommend the authors change “will” to “may” in line 82.

Line 79: Synechococcus is spelled incorrectly.

Lines 95-109: this is evidence that the GSB enriched at PHL and YNP sites may NOT be thermophilic species, particularly given the incubation temperatures of the enrichments.

Lines 129-131: Why was an incubation temperature of 67 C not tried?

Section 5 – Conclusions

Lines 148-152: I disagree with this conclusion by the authors. Multiple lines of evidence point to additional thermophilic GSB in TLS, NZ (but maybe the same strains that are already known to be present there), and that sporadic evidence suggests that thermophilic GSB are present in PHL and YNP. I believe it would be safer to present the conclusion that this study provides more information regarding the widespread distribution of GSB (mesophilic and thermophilic), but that given enrichment temperatures, and 16S profiling, only mesophilic GSB are confirmed to be in PHL and YNP, despite actual temperatures at the site at time of sampling. The authors are free to suggest that it is highly possible that thermophilic GSB are present in both of these locations, but that that has not been proven here.

Line 161: Again, I disagree here that all of these clades indicate thermophilic species.

Additional: The author, Jason M. Wood, is missing from the list of authors for the Supplementary Information.

Author Response

A major concern is whether or not to refer to the diversity we discovered as thermophilic. We believe that Wahlund et al. (1991) correctly described Cba. tepidum as a thermophile on the grounds that it was cultivated from moderately thermal springs 45-55C and could grow between 32C and 51-52C (optimally at 47-48C). We added a section to the Discussion that addresses this issue directly. It points out that, while our observations provide evidence consistent with thermophily, we lack the temperature vs growth data that would permit us to conclude thermophily. Clearly, further work would be required to resolve this.

We consider our evidence to be systematic, not sporadic. Many observations were made of many springs as conditions and opportunities permitted.

Line 87: We prefer to keep Figure 1 in the main text, as it provides biogeographical context for our study. We added a part A to the figure to emphasize the degree of geographic separation of sites we sampled. Part B illustrates how separated the three YNP thermal areas are, which would be helpful for readers unfamiliar with the features of YNP and would help guide those interested in conducting further work on these organisms.

Line 124-125: No enrichments were made in 1986 or later from the NZ mat samples. Those samples were used only for DNA extraction and sample analysis as described in text in section 2.8, lines 200-201 which has been revised to clarify

Table 1: Figure references corrected.

Section 2.3: Results are presented in text on lines 408-415 and 455-457.

Section 2.4 lines 145-149:  A sentence was added on line 150 pointing the reader to a Supplementary Methods section where full instructions for media preparation are given and a full description of when and where samples were collected.  Text lines 156-158  refer to Supplementary Information, clarify that the same medium was used for all enrichments and subcultures (the sole revival is addressed separately). Lines 158-163 address different temperatures and the use of thiosulfate in parallel enrichments and subcultures.

Line 152, 154-159: This was addressed on lines 169-173. Since this was field work, we had only limited access to incubators, so we could not test all conditions.

Lines 156-159: see response to lines 145-149. In the case of site AT, we were trying to take a broad approach to capturing possible GSB diversity at this site. We did not have an incubator available at 67C.

Lines 162-3: changed to clarify

Line 163: changed to clarify

Line 168: changed as suggested

Line 170: changed to clarify; extraction of DNA from pure cultures is described in section 2.8. Also, as stated in results, the sequencing of the genome of strain MV4-Y was done in Don Bryant’s lab and is not part of this study.

Section 2.6:         A microfuge was used at maximum speed to pellet cells. No other information was recorded as the microfuge was used only to pellet cells.

Sections 2.6, 2.7, 2.8: Text modified to further clarify.  There was only one instance in which previously frozen cells were revived and this was for the isolation of strain MV4-Y in 2011 as was specifically stated in the text. Furthermore, the text specifies that enrichments were incubated for up to 3 weeks to ensure every opportunity for growth, but in fact all enrichments that grew actually grew densely within just a few days.

Lines 221-224: Added text to speculate that GSB600F/1392R primer set may have failed in some cases because of competition of DNA from non-GSB bacteria present at the site which may have bound to the non-GSB specific 1392R primer.

Section 2.12: text modified to clarify that the phylogenetic analysis was done in 2023 from data collected in 2000 to 2002.

Figure 3: arrow added, as suggested

Sections 3.1.1 to 3.1.3:   We prefer to retain this information in Results, as site location was an important part of the study. Also, this provides a more fluid connection between sample location and enrichment and molecular results.

Line 367-368:      text changed to clarify that the periodic rising and falling was on the scale of minutes.

Lines 384-392:    text changed to clarify; no images were taken

Line 432:              no images were taken

Lines 444-446:    “Stable” changed to “viable”. Although most work on enrichments was done within 2 years, genomic sequencing was done on strain MV4-Y revived in 2011, when this technology became routine.

Lines 451-453:    We do not see the point of including a large number of spectra, when, as stated,  those displayed are representative of all and show the same results.

Figure 5:               A few arrows were added.

Section 3.5:         Yes, text modified to clarify.

Lines 488-489:    Text added to clarify that we did the genomic ANI comparison ourselves.              

Section 3.6.1:     Formatting corrected. These amplifications were done as soon as possible after collections and enrichment growth, as now stated in the Methods Section. Amplification of the TLS sample collected in 1986 and kept frozen was done in 2000, but we did not make enrichments from that sample.

Figure 6:               Floating symbols removed. We wish to keep vertical colored bars in this figure, since the blue bar shows that sequences coming from independent samples and thus colored differently belong to the same clade. The same is true in Figure 7 where the yellow bar groups sequences found in different springs with similar characteristics into a single clade.

Table 3:                Highlight removed.

Section 4—Discussion, lines 10-14:            There were no additional studies of enrichments (e.g., systematic investigation of photooxidation of sulfide/thiosulfate) other than what we mention in the text. For example, as stated in section 3.3, the spectra shown in Figure 4 are typical of the spectra from all of our GSB enrichments, such that spectra for samples shown in Figure 5 were identical to those shown in Figure 4. We do not see a need to display all spectra, given that those shown are representative. Citation of Figures 4 and 5 in Discussion was slightly revised. The information requested was added to the legend of Figure 4.

Section 4—Discussion, lines 14-18: The word “thermophilic” was removed from Section 4.1 and 4.3 titles. We added a new section that points out that, while our observations provide evidence consistent with thermophily, we lack the temperature vs growth data that would permit us to conclude thermophily. Clearly, further work would be required to resolve this. We disagree that proximity of clades in tree figures provides insight into thermophily or mesophily, as we do not know whether these traits are monophyletic. Additionally, organisms that are in differently adapted clades may be closely related.

Lines 77-78: “thermophilic” removed

Lines 79-83: Yes, this sentence summarizes previous studies described above in this paragraph, so “will” was deleted and references were provided.

Line 79: Corrected

Lines 95-109:      This should be resolved by our new Discussion section in which we consider whether the evidence is sufficient to support the inference of thermophily, which it is not. Furthermore, we do not address whether our results support mesophilic species. Both will depend on further studies.

Lines 129-131:    Limited incubators forced us to take more conservative approaches.

Lines 148-152:    We modified the Conclusions section to reflect the arguments made in the new Section mentioned above.

Line 161:              Modifications have hopefully resolved these issues.

Additional:          Thank you! His name has been added.